# Optimizing Mode Connectivity via Neuron Alignment

**N. Joseph Tatro**
Dept. of Mathematical Sciences
Rensselaer Polytechnic Institute
Troy, NY
`tatron@rpi.edu`

**Pin-Yu Chen**
IBM Research
Yorktown Heights, NY
`pin-yu.chen@ibm.com`

**Payel Das**
IBM Research
Yorktown Heights, NY
`daspa@us.ibm.com`

**Igor Melnyk**
IBM Research
Yorktown Heights, NY
`igor.melnyk@ibm.com`

**Prasanna Sattigeri**
IBM Research
Yorktown Heights, NY
`psattig@us.ibm.com`

**Rongjie Lai**
Dept. of Mathematical Sciences
Rensselaer Polytechnic Institute
Troy, NY
`lair@rpi.edu`

## Abstract

The loss landscapes of deep neural networks are not well understood due to their high nonconvexity. Empirically, the local minima of these loss functions can be connected by a learned curve in model space, along which the loss remains nearly constant; a feature known as mode connectivity. Yet, current curve finding algorithms do not consider the influence of symmetry in the loss surface created by model weight permutations. We propose a more general framework to investigate the effect of symmetry on landscape connectivity by accounting for the weight permutations of the networks being connected. To approximate the optimal permutation, we introduce an inexpensive heuristic referred to as neuron alignment. Neuron alignment promotes similarity between the distribution of intermediate activations of models along the curve. We provide theoretical analysis establishing the benefit of alignment to mode connectivity based on this simple heuristic. We empirically verify that the permutation given by alignment is locally optimal via a proximal alternating minimization scheme. Empirically, optimizing the weight permutation is critical for efficiently learning a simple, planar, low-loss curve between networks that successfully generalizes. Our alignment method can significantly alleviate the recently identified robust loss barrier on the path connecting two adversarial robust models and find more robust and accurate models on the path. Code is available at `https://github.com/IBM/NeuronAlignment`.

## 1  Introduction

Loss surfaces of neural networks have been of recent interest in the deep learning community both from a numerical (Dauphin et al., 2014; Sagun et al., 2014) and a theoretical (Choromanska et al., 2014; Safran & Shamir, 2015) perspective. Their optimization yields interesting examples of a high-dimensional non-convex problem, where counter-intuitively gradient descent methods successfully converge to non-spurious minima. Practically, recent advancements in several applications have used insights on loss surfaces to justify their approaches. For instance, Moosavi-Dezfooli et al. (2019) investigates regularizing the curvature of the loss surface to increase the robustness of trained models.

One interesting question about these non-convex loss surfaces is to what extent trained models, which correspond to local minima, are connected. Here, *connection* denotes the existence of a path between the models, parameterized by their weights, along which loss is nearly constant. There has been

conjecture that such models are connected asymptotically, with respect to the width of hidden layers. Recently, Freeman & Bruna (2016) proved this for rectified networks with one hidden layer.

When considering the connection between two neural networks, it is important for us to consider what properties of the neural networks are intrinsic. Intuitively, there is a permutation ambiguity in the indexing of units in a given hidden layer of a neural network, and as a result, this ambiguity extends to the network weights themselves. Thus, there are numerous equivalent points in model space that correspond to a given neural network, creating weight symmetry in the loss landscape. It is possible that the minimal loss paths between a network and all networks equivalent to a second network could be quite different. If we do not consider the best path among this set, we could fail to see to what extent models are intrinsically connected. Therefore, in this work we propose to develop a technique for more consistent model interpolation / optimal connection finding by investigating the effect of weight symmetry in the loss landscape. The analyses and results will give us insight into the geometry of deep neural network loss surfaces that is often hard to study theoretically.

**Related Work** Freeman & Bruna (2016) were one of the first to rigorously prove that one hidden layer rectified networks are asymptotically connected. Recent numerical works have demonstrated learning parameterized curves along which loss is nearly constant. Concurrently, Garipov et al. (2018) learned Bezier curves while Draxler et al. (2018) learned a curve using nudged elastic band energy. Gotmare et al. (2018) showed that these algorithms work for models trained using different hyperparameters. Recently, Kuditipudi et al. (2019) analyzed the connectivity between $\epsilon$-dropout stable networks. This body of work can be seen as the extension of the linear averaging of models studied in (Goodfellow et al., 2014). Recent applications of alignment include model averaging in (Singh & Jaggi, 2019) and federated learning in (Wang et al., 2018). Recently, Zhao et al. (2020) used mode connectivity to recover adversarially tampered models with limited clean data.

The symmetry groups in neural network weight space have long been formally studied (Chen et al., 1993). Ambiguity due to scaling in the weights has received much attention. Numerous regularization approaches based on weight scaling such as in (Cho & Lee, 2017) have been proposed to improve the performance of learned models. Recently, Brea et al. (2019) studied the existence of *permutation plateaus* in which the neurons in the layer of a network can all be permuted at the same cost.

A second line of work studies network similarity. Kornblith et al. (2019) gives a comprehensive review while introducing centered kernel alignment (CKA) for comparing different networks. CKA is an improvement over the CCA technique introduced in (Raghu et al., 2017) and explored further in (Morcos et al., 2018). Another contribution is the neuron alignment algorithm of (Li et al., 2016), which empirically showed that two networks of same architecture learn a subset of similar features.

**Contributions** We summarize our main contributions as follows:

1. We generalize learning a curve between two neural networks by optimizing both the permutation of the second model weights and the curve parameters. Additionally, we introduce neuron alignment to learn an approximation to the optimal permutation for *aligned* curves.

2. We demonstrate that alignment promotes similarity of the intermediate activations of models along the learned curve. Additionally, we offer analysis for a specific alignment algorithm, establishing a mechanism through which alignment improves mode connectivity.

3. We perform experiments on 3 datasets and 3 architectures affirming that more optimal curves can be learned with neuron alignment. Through utilizing a rigorous optimization method, Proximal Alternating Minimization, we observe that this aligned permutation is nearly locally optimal and consistently outperforms solutions given from random initialization.

4. For learned curves connecting adversarial robust models, we observe that the robust loss barrier recently identified in (Zhao et al., 2020) can be greatly reduced with alignment, allowing us to find more accurate robust models on the path.

## 2 Background

### 2.1 Mode Connectivity

To learn the minimal loss path connecting two neural networks, $\boldsymbol{\theta}_1$ and $\boldsymbol{\theta}_2$, with $N$ parameters, we utilize the approach of (Garipov et al., 2018). We search for the connecting path, $\boldsymbol{r} : [0, 1] \mapsto \mathbb{R}^N$,

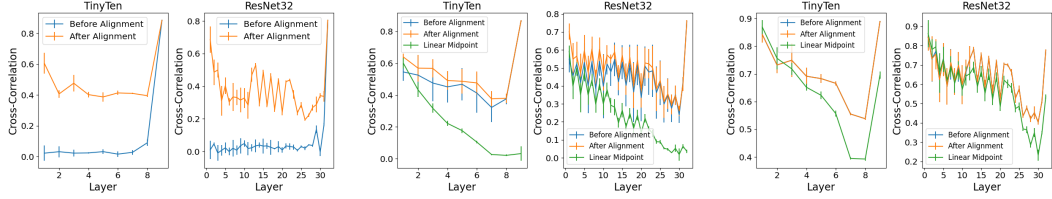

(a) Correlation between curve endpoints

(b) Correlation between endpoint and unaligned curve midpoint

(c) Correlation between endpoint and aligned curve midpoint

Figure 1: **Left:** The mean cross-correlation between corresponding units of two models for each layer before and after alignment. The standard deviation of this correlation signature over a set of different network pairs is displayed. The quality of correspondence at each layer is improved by alignment. **Middle/Right:** These plots display the correlation signatures for the endpoint models and the midpoint of the learned **unaligned/aligned** curves. This signature is shown before and after calculating the alignment of the midpoint to the endpoint model. For comparison, the signature between the endpoint and the linear initialization midpoint is shown. For the aligned curve, the orange and blue curves are almost identical. We empirically see that alignment promotes greater similarity in the intermediate activations of the midpoint model to those of the endpoint model.

that minimizes the average of the loss, $\mathcal{L}$, along the path. This problem is formalized in equation (1).

$$\boldsymbol{r}^* = \arg\min_{\boldsymbol{r}} \quad \frac{\int_{t\in[0,1]} \mathcal{L}(\boldsymbol{r}(t))\|\boldsymbol{r}'(t)\|dt}{\int_{t\in[0,1]} \|\boldsymbol{r}'(t)\|dt} \qquad \text{subject to} \quad \boldsymbol{r}(0) = \boldsymbol{\theta}_1, \boldsymbol{r}(1) = \boldsymbol{\theta}_2. \qquad (1)$$

For tractability, $\boldsymbol{r}^*$ can be approximated by a parameterized curve $\boldsymbol{r}_\phi$, where $\phi$ denotes the parameters. An arclength parameterization, $\|r'(t)\| = 1$ for all $t$, is assumed to make optimization computationally feasible. If the endpoints are global minima and a flat loss path exists, then the optimal objective of equation (1) is unchanged. Algorithm 2 in Appendix B addresses computationally solving this problem. For clarity, $r_\phi$ is the curve on the loss surface, while $\boldsymbol{r}_\phi(t)$ is a network on the curve.

## 2.2 Neuron Alignment via Assignment

Neuron alignment refers to techniques of determining a bipartite match between neurons in the same layer of different neural networks. Given input d drawn from the input data distribution $D$, let $\boldsymbol{X}_{l,i}^{(1)}(\mathrm{d}) \in \mathbb{R}^k$ represent the activations of channel $i$ in layer $l$ of network $\boldsymbol{\theta}_1$, where $k$ is the number of units in the channel. A channel could be a unit in a hidden state or a filter in a convolutional layer.

Given two networks of the same architecture, $\boldsymbol{\theta}_1$ and $\boldsymbol{\theta}_2$, we define a permutation, $\boldsymbol{P}_l : K_l \to K_l$, that maps from the index set of neurons in layer $l$ of $\boldsymbol{\theta}_1$ to $\boldsymbol{\theta}_2$. This provides a correspondence between the neurons, and we can associate a cost function $c : \mathbb{R}^k \times \mathbb{R}^k \to \mathbb{R}^+$ for the individual correspondences to minimize. This is exactly solving the *assignment problem* (Burkard & Cela, 1999),

$$\boldsymbol{P}_l^* = \arg\min_{\boldsymbol{P}_l \in \Pi_{K_l}} \sum_{i \in K_l} c(\boldsymbol{X}_{l,i}^{(1)}, \boldsymbol{X}_{l,\boldsymbol{P}_l(i)}^{(2)}). \qquad (2)$$

This has natural ties to optimal transport (Peyré et al., 2019) discussed further in Appendix C. There are many valid ways to construct $\boldsymbol{P}_l^*$ using different cost functions. For instance, in (Li et al., 2016), the cost function $c$ corresponds to $c(\boldsymbol{x},\boldsymbol{y}) := 1 - \text{correlation}(\boldsymbol{x},\boldsymbol{y})$ for $\boldsymbol{x},\boldsymbol{y} \in \mathbb{R}^k$. For clarity, correlation$(\boldsymbol{x},\boldsymbol{y})$ between channels is defined by the following equation, with channel-wise means $\boldsymbol{\mu}$ and standard deviations $\boldsymbol{\Sigma}$,

$$\text{correlation}(\boldsymbol{x},\boldsymbol{y}) = \frac{\boldsymbol{x} - \boldsymbol{\mu}_x}{\boldsymbol{\Sigma}_x}^T \frac{\boldsymbol{y} - \boldsymbol{\mu}_y}{\boldsymbol{\Sigma}_y}. \qquad (3)$$

We also note that while the method of (Li et al., 2016) uses post-activations, alignment can also be done using pre-activations. That is, $\boldsymbol{X}_{l,i}^{(1)}$ corresponds to the values at the neurons before the application of a pointwise nonlinearity, $\sigma$. Certain choices of the cost function $c$ and definition of the activations allow for more tractable theoretical analysis of alignment. Regardless, all reasonable choices promote a kind of similarity in the intermediate activations between two networks.

The correlation-based alignment is visualized in Figure 1a. This plot displays mean cross-correlation at each layer between corresponding neurons. With this *correlation signature* being increased highly with alignment, we are matching a subset of highly correlated features.

## 2.3 Adversarial Training

A recent topic of interest has been learning robust models that can withstand adversarial attacks. We specifically consider Projected Gradient Descent (PGD) attacks as described in (Madry et al., 2017). This is an evasion attack that adds optimized $l_\infty$ bounded noise to input to degrade accuracy. Security to these are important as they can be conducted without access to model parameters as in (Chen et al., 2017). Moreover, adversarial attacks can be used during model training to improve adversarial robustness, a method known as adversarial training (Goodfellow et al., 2015; Madry et al., 2017).

## 3  Mode Connectivity with Weight Symmetry

We clarify the idea of weight symmetry in a neural network. Let $\boldsymbol{\theta}_1$ be a network on the loss surface parameterized by its weights. A permutation $\boldsymbol{P}_l$ is in $\Pi_{|K_l|}$, the set of permutations on the index set of channels in layer $l$. For simplicity suppose we have an $L$ layer feed-forward network with pointwise activation function $\sigma$, weights $\{W_l\}_{l=1}^L$, and input $X_0$. Then the weight permutation ambiguity becomes clear when we introduce the following set of permutations to the feedforward equation:

$$\boldsymbol{Y} := \boldsymbol{W}_L \boldsymbol{P}_{L-1}^T \circ \sigma \circ \boldsymbol{P}_{L-1} \boldsymbol{W}_{L-1} \boldsymbol{P}_{L-2}^T \circ \ldots \circ \sigma \circ \boldsymbol{P}_1 \boldsymbol{W}_1 \boldsymbol{X}_0. \tag{4}$$

Then we can define the network weight permutation $\boldsymbol{P}$ as the block diagonal matrix, blockdiag($\boldsymbol{P}_1, \boldsymbol{P}_2, ..., \boldsymbol{P}_{L-1}$). Additionally, $\boldsymbol{P}\boldsymbol{\theta}$ denotes the network parameterized by the weights $[\boldsymbol{P}_1 \boldsymbol{W}_1, \boldsymbol{P}_2 \boldsymbol{W}_2 \boldsymbol{P}_1^T, ..., \boldsymbol{W}_L \boldsymbol{P}_{L-1}^T]$. Note that we omit permutations $\boldsymbol{P}_0$ and $\boldsymbol{P}_L$, as the input and output channels of neural networks have a fixed ordering. It is critical that $\boldsymbol{P}_i$ is a permutation, as more general linear transformations do not commute with $\sigma$. Without much difficulty this framework generalizes for more complicated architectures. We discuss this for residual networks in Appendix D.

From equation (4), it becomes clear that the networks $\boldsymbol{\theta}_1$ and $\boldsymbol{P}\boldsymbol{\theta}_1$ share the same structure and intermediate outputs up to indexing. Taking weight symmetry into account, we can find the optimal curve connecting two networks up to symmetry with the model in equation (5).

$$\min_{\phi, \boldsymbol{P}} \quad \mathbb{E}_t[\mathcal{L}(\boldsymbol{r}_\phi(t))] \quad s.t. \quad \boldsymbol{r}_\phi(0) = \boldsymbol{\theta}_1, \boldsymbol{r}_\phi(1) = \boldsymbol{P}\boldsymbol{\theta}_2, \boldsymbol{P} = \text{blockdiag}(\{\boldsymbol{P}_i \in \Pi_{|K_i|}\}_{i=1}^{L-1}). \tag{5}$$

### 3.1  Neuron Alignment for Approximating the Optimal Permutation

With the correlation signatures of Figure 1, it is clear that trained networks of the same architecture share some similar structure. Then we could expect that if a model along the curve, $\boldsymbol{r}(t)$, is optimal, it shares some structure of $\boldsymbol{r}(0)$ and $\boldsymbol{r}(1)$. To this end, it is sensible that the curve connecting the models interpolates weights of similar filters. This motivates our use of neuron alignment to estimate $\boldsymbol{P}^*$ in equation (5), so that we are learning a curve between *aligned networks*. In Section 4, we confirm increased similarity of intermediate activations of models along the curve to endpoint models.

**Theory for Using Neuron Alignment**   We can theoretically establish how increased similarity of intermediate activation distributions benefit mode connectivity for a specific method of alignment. Namely, we consider an alignment of the neurons minimizing the expected $L_2$ distance between their pre-activations. Then this permutation is an optimal transport plan associated with the Wasserstein-2 metric, discussed further in Appendix C. The following theorem derives tighter bounds on the loss of the linear interpolations between models after alignment compared to before alignment. Then these upper bounds extend to the optimal curves themselves.

Let the neural networks $\boldsymbol{\theta}$ have a simple feed-forward structure as in equation (4) with Lipschitz-continuous $\sigma$. The loss function $\mathcal{L}$ is also assumed to be Lipschitz-continuous or continuous with the input data distribution bounded. The function, $F$, is taken to denote the objective value of equation (5) given curve parameters and permutation.

**Theorem 3.1** (Alignment Leads to Tighter Loss Bound)**.** *Let the above assumptions be met. Consider the following solutions to equation (5), $(\phi_u^*, \mathbf{I})$ and $(\phi_a^*, \boldsymbol{P})$, where the corresponding curve*

*parameters, φ, are optimal for the given permutation. Here $\boldsymbol{P}$ is the solution to equation (2) using the aforementioned alignment technique. Then there exists upper bounds $B_u$ and $B_a$ such that*

$$F(\phi_u^*, \mathbf{I}) \leq B_u, \quad F(\phi_a^*, \boldsymbol{P}) \leq B_a \quad where \quad B_a \leq B_u. \tag{6}$$

*Proof.* See Appendix C.1 for the complete proof. We provide an outline below.

Consider the model at point $t$ along the linear interpolation between two endpoint models. It follows that the preactivations of the first layer of this model are closer (in the $W_2$ sense) to those of an endpoint after alignment than before alignment. After applying the nonlinear activation function, we have a tighter upper bound on the distance between the first layer activations of the model and those of an endpoint in the aligned case via Lipschitz continuity. Using matrix norm inequalities of the layer weights, we find a tighter bound on the distance between the preactivations of the second layer in the model and those of an endpoint model after alignment.

We iteratively repeat this process to find a tighter upper bound on the distance between the model output to those of an endpoint model after alignment. Finally, exploiting the Lipschitz continuity of the restricted loss function, we have a tighter bound on the loss of the model output after alignment. As a tighter bound can be found at each point along the linear interpolation for aligned models, it provides a tighter bound on the associated average loss. This tighter upper bound for the linear interpolation from using alignment is then clearly a tighter upper bound for the optimal curve. □

The difficulty in directly comparing $F(\phi_u^*, \mathbf{I})$ and $F(\phi_a^*, \boldsymbol{P})$ stems from the nonlinear activation function. Thus, we bound each of these values individually. We now discuss tightness of these bounds. In the Appendix C.2, we explore what conditions need to be met for the bounds on the average loss of the linear interpolations to be tight. Given the bounds for loss along linear interpolations are tight, we can consider the tightness of the bound for loss along piecewise linear curves between networks. These piecewise linear curves can themselves approximate continuous curves. Given a piecewise linear curve with optimal loss, for two models on the same line segment of this curve, we can consider the optimal piecewise linear curve between them. It follows that this optimal curve is their linear interpolation. Therefore, we have tightness in the upper bound for curve-finding restricted to piecewise linear curves between networks of a class for which we have tightness for the linear interpolation. Via an argument by continuity, this tightness extends to continuous curves.

Empirically, alignment using the correlation of post-activations outperforms that of alignment associated with the Wasserstein-2 metric, as used in (Arjovsky et al., 2017), for pre-activations. See Figure 2. We discuss this in Appendix C.3. This observation motivates our use of the former technique throughout this paper. Additionally, cross-correlation is often preferred over unnormalized signals in certain tasks as in (Avants et al., 2008). The different techniques are still fundamentally related.

**Numerical Implementation of Neuron Alignment** Algorithm 1 details computing a permutation of the network weights from neuron alignment. This demonstrates a correlation-based alignment of post-activations, though it easily generalizes for other techniques. In practice, we solve the assignment problem in equation (2) using the Hungarian algorithm (Kuhn, 1955). For an $L$ layer network with maximum width of $M$, we compute $\boldsymbol{P}$ using a subset of the training data. Then the cost of computing the cross-correlation matrices for all layers is dominated by the forward propogation through the network to accumulate the activations. The running time needed to compute all needed assignments

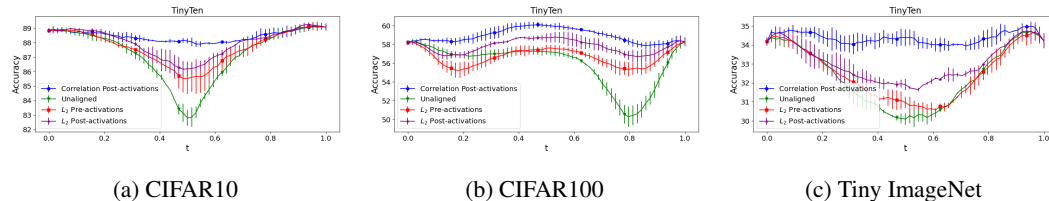

(a) CIFAR10        (b) CIFAR100        (c) Tiny ImageNet

Figure 2: Comparison of different alignment techniques across the different datasets for the TinyTen architecture. These plots show the test accuracy along the learned curves. Notice that all methods outperform the *Unaligned* case. Notably, alignment of post-activations outperforms alignment of pre-activations. Maximizing cross-correlation clearly outperforms minimizing $L_2$ distance.

**Algorithm 1:** Permutation via Neuron Alignment

---

**Input:** Trained Neural Networks $\boldsymbol{\theta}_1$ and $\boldsymbol{\theta}_2$, Subset of Training Data $\mathbf{X}_0$
**Output:** Aligned Neural Networks $\boldsymbol{\theta}_1$ and $\boldsymbol{P}\boldsymbol{\theta}_2$
Initialize $\boldsymbol{P}\boldsymbol{\theta}_2$ as $[\boldsymbol{W}_1^2, \boldsymbol{W}_2^2, \dots, \boldsymbol{W}_L^2]$
**for** each layer $l$ in $\{1, 2, \dots, L-1\}$ **do**
    **for** each network $j$ in $\{1, 2\}$ **do**
        compute activations, $\mathbf{X}_l^{(j)} = \sigma \circ \boldsymbol{W}_l^j \mathbf{X}_{l-1}^{(j)}$
        compute, $\boldsymbol{Z}_l^{(j)}$, the normalization of the vectorized activations
    compute cross-correlation matrix, $\boldsymbol{C}_l^{(1,2)} = \boldsymbol{Z}_l^{(1)} \boldsymbol{Z}_l^{(2)T}$
    compute $\boldsymbol{P}_l$ by solving the assignment problem in eq. 2 with $\boldsymbol{C}_l^{(1,2)}$ using Hungarian algorithm
    update $\hat{\boldsymbol{W}}_l^2 \rightarrow \boldsymbol{P}_l \hat{\boldsymbol{W}}_l^2, \quad \hat{\boldsymbol{W}}_{l+1}^2 \rightarrow \hat{\boldsymbol{W}}_{l+1}^2 \boldsymbol{P}_l^T$

---

is $\mathcal{O}(LM^3)$, with storage $\mathcal{O}(LM)$. This is on the order of the running time associated with one iteration of forward propagation. Then neuron alignment is relatively cheap as the time complexity of computing aligned curves is on the same order as traditional curve finding. We also stress that in practice only a subset of training data is needed to compute alignment. We confirmed that the computed alignment for models trained on CIFAR100 was the same given subset sizes of (2,500, 5,000, and 10,000). This choice of a smaller subset leads to faster computation.

**On Rigorously Learning the Permutation in the Joint Model** We explore rigorously learning a more optimal permutation using a Proximal Alternating Minimization (PAM) scheme (Attouch et al., 2010). The PAM scheme, formalized the first equation of Appendix E, solves equation (5) via alternatively optimizing the weight permutation, $\boldsymbol{P}$, and the curve parameters, $\phi$, with added proximal penalty terms. Under some assumptions, PAM guarantees global convergence to a local minima, meaning the quality of the solution is tied to the initialization. As such, PAM provides a baseline for establishing if a solution to equation (5) is locally optimal.

We implement and solve a PAM scheme starting from both the unaligned and aligned permutation initializations to gain insight into the optimality of the learned aligned curve. We establish theoretical guarantees and discuss the numerical implementation of PAM in Appendix E.

**Using Alignment to Connect Adversarial Robust Models** So far, we have discussed aligning the features of two typical neural networks to benefit learning a low loss curve connecting said networks. We are also interested in learning such curves between adversarially robust networks. This is motivated by recent observations in (Zhao et al., 2020) that found the existence of a loss barrier between adversarial robust networks. In Section 5, we discuss our findings that such a barrier is in large part due to artifacts of weight permutation symmetry in the loss landscape.

## 4 Experiments

**Datasets** We trained neural networks to classify images from CIFAR10 and CIFAR100 (Krizhevsky et al., 2009), as well as Tiny ImageNet (Deng et al., 2009). The default training and test set splits are used for each dataset. Cross entropy loss of the output logits is used as the loss function. $20\%$ of the images in the training set are used for computing alignments between pairs of models. We augment the data using color normalization, random horizontal flips, random rotation, and random cropping.

**Architectures** Three different model architectures are used. They are included in Table 1 along with their number of parameters. The first architecture considered is the TinyTen model. TinyTen, introduced in (Kornblith et al., 2019), is a narrow 10 layer convolutional neural network. This is a useful model for concept testing and allows us to gain insight to networks that are underparameterized. We also include ResNet32 (He et al., 2016) in our experiments to understand the effect of skip connections. Finally, we consider the GoogLeNet architecture, which is significantly wider than the previously mentioned networks and achieves higher performance (Szegedy et al., 2015).

Table 1: Average accuracy along the curve with standard deviation is reported for each combination of dataset, network architecture, and curve class. For emphasis, we list the performance of the worst model found along the curve. The average accuracy of the endpoint models are also included. Aligned curves clearly outperform the unaligned curves of (Garipov et al., 2018). Number of parameters is also included.

| Model | Average/Minimum Accuracy of Models Along the Learned Curve | | | | | |
| --- | --- | --- | --- | --- | --- | --- |
| | CIFAR10 | | CIFAR100 | | Tiny ImageNet | |
| TinyTen (0.09M) | $89.0 \pm 0.1$ | | $58.1 \pm 0.5$ | | $34.2 \pm 0.2$ | |
| Unaligned (Garipov) | $87.4 \pm 0.1$ | $82.8 \pm 0.5$ | $56.0 \pm 0.2$ | $53.2 \pm 1.1$ | $32.5 \pm 0.1$ | $30.0 \pm 0.3$ |
| PAM Unaligned | $87.6 \pm 0.1$ | $84.0 \pm 0.3$ | $57.3 \pm 0.2$ | $55.9 \pm 0.9$ | $33.6 \pm 0.1$ | $32.5 \pm 0.1$ |
| PAM Aligned | $88.4 \pm 0.1$ | $87.6 \pm 0.2$ | $\mathbf{58.8 \pm 0.1}$ | $\mathbf{57.7 \pm 0.3}$ | $34.2 \pm 0.1$ | $33.4 \pm 0.2$ |
| Aligned | $\mathbf{88.5 \pm 0.1}$ | $\mathbf{87.8 \pm 0.1}$ | $58.7 \pm 0.2$ | $57.7 \pm 0.4$ | $\mathbf{34.4 \pm 0.1}$ | $\mathbf{33.7 \pm 0.1}$ |
| ResNet32 (0.47M) | $92.9 \pm 0.1$ | | $67.1 \pm 0.5$ | | $50.2 \pm 0.0$ | |
| Unaligned (Garipov) | $92.2 \pm 0.1$ | $89.1 \pm 0.2$ | $66.5 \pm 0.2$ | $64.7 \pm 0.4$ | $48.2 \pm 0.1$ | $45.2 \pm 0.1$ |
| PAM Unaligned | $92.4 \pm 0.1$ | $89.9 \pm 0.2$ | $67.0 \pm 0.1$ | $66.1 \pm 0.1$ | $48.5 \pm 0.1$ | $46.6 \pm 0.1$ |
| PAM Aligned | $\mathbf{92.7 \pm 0.0}$ | $92.1 \pm 0.1$ | $67.6 \pm 0.4$ | $\mathbf{66.8 \pm 0.1}$ | $49.2 \pm 0.4$ | $47.9 \pm 0.6$ |
| Aligned | $\mathbf{92.7 \pm 0.1}$ | $\mathbf{92.2 \pm 0.0}$ | $\mathbf{67.7 \pm 0.1}$ | $66.6 \pm 0.1$ | $\mathbf{49.5 \pm 0.3}$ | $\mathbf{48.8 \pm 0.4}$ |
| GoogLeNet (10.24M) | $93.4 \pm 0.0$ | | $73.2 \pm 0.4$ | | $51.6 \pm 0.2$ | |
| Unaligned (Garipov) | $93.3 \pm 0.0$ | $92.1 \pm 0.1$ | $73.1 \pm 0.4$ | $69.8 \pm 0.3$ | $51.9 \pm 0.1$ | $48.7 \pm 0.4$ |
| Aligned | $\mathbf{93.4 \pm 0.0}$ | $\mathbf{93.1 \pm 0.0}$ | $\mathbf{73.4 \pm 0.3}$ | $\mathbf{72.9 \pm 0.3}$ | $\mathbf{52.4 \pm 0.2}$ | $\mathbf{51.4 \pm 0.3}$ |

All models used as curve endpoints are trained using SGD. We set a learning rate of $1\mathrm{E}{-}1$ that decays by a factor of $0.5$ every 20 epochs. Weight decay of $5\mathrm{E}{-}4$ was used for regularization. Each model was trained for 250 epochs, and all models were seen to converge. This training scheme produced models of comparable accuracy to those in related literature such as (Kornblith et al., 2019). Models were trained on NVIDIA 2080 Ti GPUs.

**Training curves** All curves are parameterized as quadratic Bezier curves. Bezier curves are defined by their *control points*. In the current study, we refer to endpoint models as $\boldsymbol{\theta}_1$ and $\boldsymbol{\theta}_2$ as well as the control point, $\boldsymbol{\theta}_c$. Then $\boldsymbol{r}$ is defined in equation (7) with $\boldsymbol{\theta}_c$ as the learnable parameter in $\phi$

$$\boldsymbol{r}_\phi(t) = (1-t)^2\boldsymbol{\theta}_1 + 2(1-t)t\boldsymbol{\theta}_c + t^2\boldsymbol{\theta}_2. \tag{7}$$

For each architecture and dataset, we train 6 models using different random initializations. Thus we have 3 independent model pairs. We learn two classes of curves, *Unaligned / Aligned*, that are solutions to algorithm 2 for $\boldsymbol{\theta}_1$ and $\boldsymbol{\theta}_2$ / $\boldsymbol{P}\boldsymbol{\theta}_2$. Here $\boldsymbol{P}$ is the permutation learned by alignment. We also learn the corresponding curve classes, *PAM Unaligned / PAM Aligned*, that are solutions to the first equation in Appendix E. Their permutation intializations are $\boldsymbol{I}$ and $\boldsymbol{P}$ respectively. Tables are generated with curves trained from different random seeds, while figures are generated with curves trained from the same seed. Curves trained from different seeds are similar up to symmetry, but using them to generate the figures would average out shared features.

Curves are trained for 250 epochs using SGD with a learning rate of $1\mathrm{E}{-}2$ and a batch size of 128. The rate anneals by a factor of $0.5$ every 20 epochs. The initial hyperparameters were chosen to match those in (Garipov et al., 2018). For CIFAR100 curves, this learning rate was set to $1\mathrm{E}{-}1$, as they were seen to perform marginally better. For the TinyTen/CIFAR100 case, we explore the hyperparameter space in Appendix F to show that our results hold under different choice of hyperparameters.

### 4.1 Results on using Neuron Alignment

**Aligned Curves Outperform Unaligned Curves** The test accuracy of learned curves can be seen for each dataset, architecture, and curve class in Table 1. Clearly, the aligned curves outperform the unaligned curve. In many cases, the average accuracy along the aligned curves in comparable to the trained models used as endpoints. The table also contains the minimum accuracy along the curve, indicating that aligned curves do not suffer from the same generalization gap that unaligned curves are prone to. Finally, Table 2 in the appendix contains the training loss for each case at convergence. Overall, the strongest gains from using alignment are for underparameterized networks. As seen in Table 1, the largest increase in performance is for TinyTen on Tiny ImageNet while the smallest gain is made for GoogLeNet on CIFAR10. This is inline with observations by (Freeman & Bruna, 2016).

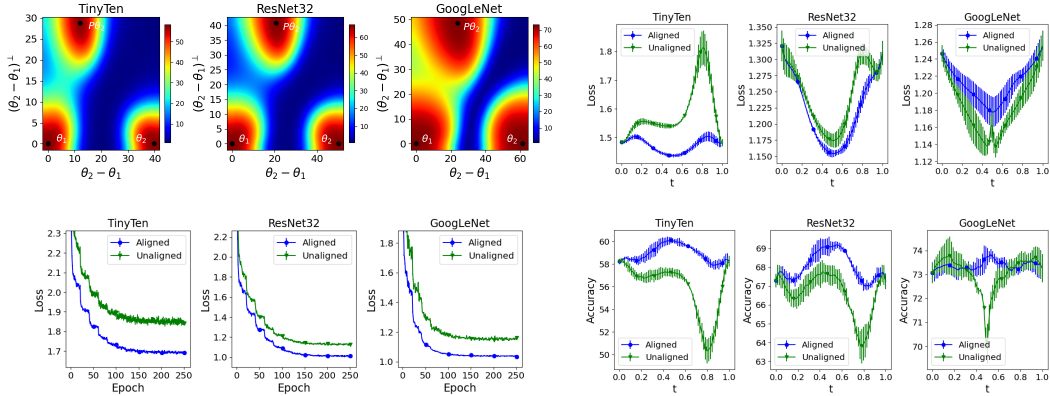

Figure 3: **Top Left:** Test accuracy on CIFAR100 across the plane containing $\boldsymbol{\theta}_1$, $\boldsymbol{\theta}_2$, and $\boldsymbol{P}\boldsymbol{\theta}_2$. This plane contains the two different intializations used in our curve finding experiments. The default initialization, $\boldsymbol{\theta}_2 - \boldsymbol{\theta}_1$, and the aligned initialization, $\boldsymbol{P}\boldsymbol{\theta}_2 - \boldsymbol{\theta}_1$. This shows that the aligned initialization is notably better. **Bottom Left**: Training loss during training. **Top/Bottom Right:** Test loss/accuracy along these curves. Aligned curves generalize better and do not suffer from large drops in accuracy typical for unaligned curves.

The test loss and accuracy along the learned curves for CIFAR100 are shown in Figure 3. We observe that, as expected, the accuracy at each point along the aligned curve usually exceeds that of the unaligned curve, while the loss along the curve is also smoother with neuron alignment. Of note is the prominent presence of an accuracy barrier along the unaligned curves.

Figure 3 displays the planes containing the initializations for curve finding. Clearly the aligned initialization has better objective value. The axis is determined by Gram-Schmidt orthonormalization. In Appendix A the planes containing the learned curves are displayed in Figures 7. The loss displayed on the planes containing the linear initializations and the curves can be seen in Figure 5.

Practically, using neuron alignment for determining the permutation $\boldsymbol{P}$ may be enough and avoids more complicated optimization. Note the relative flatness of the accuracy along the aligned curves in Figure 3. Additionally, the plots in the top row indicate much faster convergence when learning $\phi$ using neuron alignment. For example, the aligned curve takes 100 epochs less to achieve the training accuracy that the unaligned curve converges to, for TinyTen and CIFAR100. Figures 6 and 7 in Appendix A displays the previously mentioned plots for the additional datasets.

**Similarity of Intermediate Activations Along the Curve**   In Figures 1b and 1c, the plots display the similarity of intermediate activations between the learned curve midpoints and the endpoint models. This is evidence that alignment increases this similarity. Notice that even when the endpoints are not aligned, the learned midpoint on the unaligned curve is seen to be mostly aligned to the endpoints. Thus, the similarity of these distributions is important to mode connectivity, and we can take the view that these learned curves are trying to smoothly interpolate features.

**Insights on Local Optimality via PAM**   As seen in Table 1, starting from the unaligned initialization, it is possible to find a better permutation for optimizing mode connectivity. In contrast, the permutation given by alignment is nearly locally optimal as *PAM Aligned* and *Aligned* are comparable. Even more, the solutions to PAM with an unaligned initialization, *PAM*, still do not perform better than curve finding with the aligned permutation, *Aligned*. This implies that the neuron alignment heuristic is not only inexpensive, it provides a permutation of optimality not likely found without conducting some kind of intractable search. We see this as a strength of neuron alignment.

## 5   Mode Connectivity of Adversarial Robust Models

**Alignment greatly reduces the robust loss barrier**   Figure 4 displays the training loss and test accuracy of the learned robust curve between adversarially trained robust CIFAR100 models for the previously mentioned architectures. The left plots displays standard test accuracy whereas the right

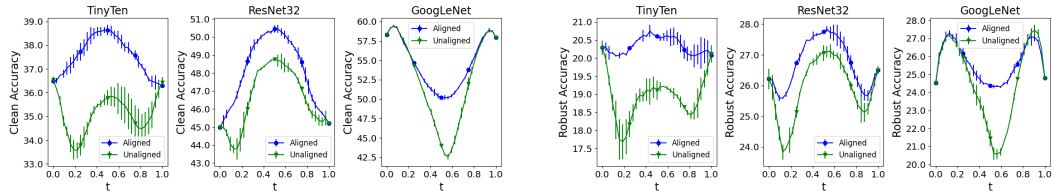

Figure 4: **Left/Right**: Clean/Robust test accuracy along these curves. For TinyTen and ResNet32, it is clear that a more robust and accurate model can be found along the curve compared to the endpoints. GoogLeNet results are more complicated due to well known overfitting of overparameterized models during adversarial training (Rice et al., 2020).

plots displays the test accuracy of PGD adversarial examples. These networks and curves are trained with the same scheme as in (Zhao et al., 2020), where we set the initial learning rate to $1E-1$. An important point to emphasize is that the curve itself is trained to minimize robust loss, so the input undergoes PGD attack at each point along the curve.

For the robust curve learned between two unaligned robust models, we encounter barriers both in clean and robust accuracy, as reported in (Zhao et al., 2020). As in Figure 3, these accuracy barriers appear to correspond with barriers in loss, where plots of robust loss along these curves can be found in Figure 8 in Appendix A. It is clear that the barrier in clean accuracy is eliminated or greatly reduced with the use of alignment. With respect to robust accuracy, we see that alignment significantly alleviates that barrier. In practice, adversarial training of the GoogLeNet curves were found to overfit on the training data. This has been well observed for over-parameterized models (Rice et al., 2020). Thus, the curves displayed for GoogLeNet are the results with early stopping after roughly 20 epochs to prevent overfitting. Figure 9 in the Appendix A displays these results for CIFAR10.

**Aligned curves can find more accurate robust models.** Neuron alignment seems successful at finding a curve between robust models along which models maintain their robustness to PGD attacks without sacrificing clean accuracy. Results provide evidence that the presence of a large robust loss barrier between robust models as mentioned in (Zhao et al., 2020) can mostly be attributed as an artifact of symmetry in the loss landscape resulting from the network weights.

For CIFAR100, alignment enables finding a more accurate model without sacrificing robust accuracy, which provides new insight towards overcoming the issue of robustness-accuracy tradeoff in adversarial robustness (Su et al., 2018). Consider the midpoint on the ResNet32 aligned curve in Figure 4, where both clean and robust accuracies increase by $5.3\%$ and $1.3\%$, respectively, in comparison to the endpoints. For TinyTen, these accuracies also increase at the aligned curve midpoint, while no better model in term of clean or robust accuracy exists along the unaligned curve with respect to the endpoints. For GoogLeNet, we find comparable more accurate and robust models near the endpoints for both curves, though only the aligned curve avoids a robust accuracy barrier. We emphasize that learning a better model from scratch is not an easy task. In practice, converging here requires a step size large enough for the SGD trajectory to reach this basin within feasible time and that is then small enough to converge in the basin. Thus, the aligned curve finding can be viewed as a technique for avoiding hyperparameter tuning, which is typically expensive.

## 6 Conclusion

We generalize mode connectivity by removing the weight symmetry ambiguity associated with the endpoint models. The optimal permutation of these weights can be approximated using neuron alignment. We empirically find that this approximation is locally optimal and outperforms the locally optimal solution to a proximal alternating scheme with random initialization. Empirically, we show that neuron alignment can be used to successfully and efficiently learn optimal connections between neural nets. Addressing symmetry is critical for learning planar curves on the loss surface along which accuracy is mostly constant. Our results hold true over a range of datasets and network architectures. With neuron alignment, these curves can be trained in less epochs and to higher accuracy. Novel to previous findings, with alignment we also find that robust models are in fact connected on the loss surface and curve finding serves as a means to identify more accurate robust models.

## Broader Impact

This work examines solving the problem of mode connectivity up to a symmetry in the weights of the given models. Our method allows for the computation of a curve of nearly optimal models, where this curve itself has a simple parameterization. We discuss the broader impacts of this work from the following perspectives:

- **Who may benefit from this research:** In this work we show the ability to learn simply parameterized curves along which each model is seen to be robust. This could have potential applications in ensembling, where an ensemble of robust networks can be learned without training each individual model. Generally, an attack on an ensemble will be less effective than on a direct attack on any of its component models. Additionally, regarding CIFAR100, we see the ability to find comparable robust models of greater accuracy along the curve. Thus, this work can benefit systems for which robustness is critical.

- **Who may be disadvantaged from this research:** As mentioned, this work can benefit systems for which robustness is critical. Such systems are typically part of a movement towards *trusted artificial intelligence*. As trust in systems increases, these systems may see wider use and adoption, such as self-driving cars. With increased automation, workers such as truck drivers stand to have declining career prospects. Thus, this work is part of a broader push in research that may disadvantage these peoples.

- **Consequences of failure:** If our method fails for a given instance, then it means that we were unable to learn a simple curve along which the models are nearly optimal. This means the method cannot be used in that instance for an application such as providing a set of models for ensembling. In the case of adversarial models, this means that the learned models are vulnerable to attacks and becoming compromised.

- **Biases in the data:** In our experiments, we validated our results for three different datasets to confirm that our method does not depend on a bias uniquely associated with an individual dataset.

## Acknowledgments and Disclosure of Funding

This work was supported by the Rensselaer-IBM AI Research Collaboration (http://airc.rpi.edu), part of the IBM AI Horizons Network (http://ibm.biz/AIHorizons). Additionally, R. Lai's work is supported in part by NSF CAREER Award (DMS—1752934). The authors also thank Youssef Mroueh for helpful discussions on optimal transport.

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
