[Supplementary Material · Optimizing_Mode_Connectivity_via_Neuron_Alignment_Supplementary.pdf]

# A  Additional Figures

(a) CIFAR100

(b) CIFAR10

(d) CIFAR10

(c) Tiny ImageNet

(e) Tiny ImageNet

Figure 5: **Left/Right:** Test loss/accuracy on plane containing $\theta_1$, $\theta_2$, and $P\theta_2$.

Figure 6: **Left/Right**: Training loss while learning the curve between two CIFAR10/Tiny ImageNet models.

Table 2: The training loss with standard deviation is reported for each combination of dataset, network architecture, and curve class. GoogLeNet has higher training loss due to weight regularization.

| Model | Endpoints | CIFAR10 | CIFAR100 | Tiny ImageNet |
|---|---|---|---|---|
| TinyTen | Unaligned | $0.428 \pm 0.003$ | $1.839 \pm 0.010$ | $3.317 \pm 0.008$ |
| | PAM Unaligned | $0.413 \pm 0.001$ | $1.753 \pm 0.016$ | $3.249 \pm 0.005$ |
| | PAM Aligned | $0.372 \pm 0.002$ | $\mathbf{1.679 \pm 0.005}$ | $\mathbf{3.214 \pm 0.003}$ |
| | Aligned | $\mathbf{0.371 \pm 0.002}$ | $1.693 \pm 0.008$ | $3.217 \pm 0.013$ |
| ResNet32 | Unaligned | $0.179 \pm 0.001$ | $1.124 \pm 0.005$ | $2.383 \pm 0.005$ |
| | PAM Unaligned | $0.170 \pm 0.001$ | $1.043 \pm 0.008$ | $2.350 \pm 0.001$ |
| | PAM Aligned | $\mathbf{0.147 \pm 0.002}$ | $\mathbf{0.975 \pm 0.008}$ | $2.308 \pm 0.003$ |
| | Aligned | $\mathbf{0.147 \pm 0.001}$ | $1.011 \pm 0.002$ | $\mathbf{2.299 \pm 0.009}$ |
| GoogLeNet | Unaligned | $0.540 \pm 0.001$ | $1.161 \pm 0.004$ | $2.570 \pm 0.009$ |
| | Aligned | $\mathbf{0.516 \pm 0.001}$ | $\mathbf{1.033 \pm 0.002}$ | $\mathbf{2.278 \pm 0.005}$ |

# B  Algorithms

This section contains algorithms described in Section 2. In the curve finding algorithm, the optimization step can correspond to a variety of techniques. In this paper, we use traditional stochastic

(a) CIFAR100

(d) CIFAR100

(b) CIFAR10

(e) CIFAR10

(c) Tiny ImageNet

(f) Tiny ImageNet

Figure 7: **Left/Right:** The test loss/accuracy on plane containing learned curve, $r_\phi(t)$.

gradient descent to update the curve parameters $\phi$. Notice that stochasticity is introduced by the sampling of $t$ as well as the training data. This is detailed in Algorithm 2.

For the purpose of computing validation loss and test loss for $r_\phi$, important care must be given for networks that contain batch normalization layers. This is because batch normalization aggregates running statistics of the network output that are used when evaluating the model. Though, $r_\phi(t_0)$ gives the weights for the model at point $t_0$, the running statistics need to be aggregated for each normalization layer. In practice, this can be done by training the model for one epoch, while freezing all learnable parameters of the model. Since batch statistics would need to be computed for each point sampled along the curve, it happens that computing the validation or test loss of the curve $r_\phi$ is more expensive than an epoch of training.

## C   Theoretical Motivation for Mode Connectivity with Neuron Alignment

In this section we present a theoretical discussion regarding the use of neuron alignment for curve finding up to weight symmetry. We begin by defining relevant terminology.

(a) CIFAR100          (b) CIFAR10

Figure 8: Robust test loss on curve between robust models.

(a) CIFAR10 Clean Accuracy       (b) CIFAR10 Robust Accuracy

Figure 9: Clean/Robust accuracy on the CIFAR10 robust curves.

**Wasserstein distance**   In the following proof, we will make use of the Wasserstein-2 metric for measuring a distance between probability distributions. This metric has recently been popular in works such as WGAN (Arjovsky et al., 2017). Formally, let $\mu$ and $\nu$ be probability measures, i.e. distributions, on a given metric space $M$. We let $\Gamma(\mu, \nu)$ denote the set of joint probability measures with marginals $\mu$ and $\nu$. Then the Wasserstein-p metric is defined as

$$W_p(\mu, \nu) := \left( \inf_{\gamma \in \Gamma(\nu, \mu)} \mathbb{E}_{(\mathbf{x}, \mathbf{y}) \sim \gamma}[||\mathbf{x} - \mathbf{y}||^p] \right)^{\frac{1}{p}} \tag{8}$$

Note that this metric is related to optimal transport, where $\gamma^*$ is the optimal transport plan between the two distributions with the cost being the Euclidean p-norm. Remember that the distributions we are interested in our the distributions of intermediate activations of neuron networks, $\boldsymbol{X}_l^{(1)}$ and $\boldsymbol{X}_l^{(2)}$, as discussed in 2.2. Then equation (8) simplifies to

$$W_p(\boldsymbol{X}_l^{(1)}, \boldsymbol{X}_l^{(2)})^p = \min_{\boldsymbol{P} \in \Pi_{K_l}} \sum_{i \in K_l} ||\boldsymbol{X}_{l,i}^{(1)} - \boldsymbol{X}_{l,\boldsymbol{P}(i)}^{(1)}||^2. \tag{9}$$

This comes from the fact that since we are dealing with an empirical distribution with uniform marginals, so the set of permutations extremal points and thus contains a minimizer.

### C.1   Proof of Theorem 3.1

For this proof, we consider a pair of feed-forward networks, with output defined as

$$\boldsymbol{Y}_j = \boldsymbol{W}_L^{(j)} \sigma \boldsymbol{W}_{L-1}^{(j)} \dots \sigma \boldsymbol{W}_1^{(j)} \boldsymbol{X}_0, \quad i = 1, 2, \tag{10}$$

---

**Algorithm 2:** Curve Finding (Garipov et al., 2018)

---

**Input:** Two trained models, $\boldsymbol{\theta}_1$ and $\boldsymbol{\theta}_2$
**Output:** A parameterized curve, $\boldsymbol{r}_\phi$, connecting $\boldsymbol{\theta}_1$ and $\boldsymbol{\theta}_2$ along which loss is flat
Initialize $\boldsymbol{r}_\phi(t)$ as $\boldsymbol{\theta}_1 + t(\boldsymbol{\theta}_2 - \boldsymbol{\theta}_1)$;
**while** not converged **do**
    **for** batch in dataset **do**
        sample point $t_0$ in $[0, 1]$
        compute loss $L(\boldsymbol{r}_\phi(t_0))$
        optimization step on network $\boldsymbol{r}_\phi(t_0)$ to update $\phi$

---

for the given input data distribution $X_0$. Now we consider the addition of the permutation matrices, $\boldsymbol{P}_i$, that generalize the above equation to deal with weight symmetry,

$$\boldsymbol{Y}_j = \boldsymbol{W}_L^{(j)} \boldsymbol{P}_{L-1}^T \sigma \boldsymbol{P}_{L-1} \boldsymbol{W}_{L-1}^{(j)} \boldsymbol{P}_{L-2}^T \dots \boldsymbol{P}_1^T \sigma \boldsymbol{P}_1 \boldsymbol{W}_1^{(j)} \boldsymbol{X}_0. \tag{11}$$

The initialization used for curve finding is the interpolation between the two neural networks. This allows us to define the unaligned and aligned linear interpolations,

$$\boldsymbol{l}_u(t) = \left((1-t)\boldsymbol{W}_L^{(1)} + t\boldsymbol{W}_L^{(2)}\right) \sigma \left((1-t)\boldsymbol{W}_{L-1}^{(1)} + t\boldsymbol{W}_{L-1}^{(2)}\right) \dots \tag{12}$$
$$\sigma \left((1-t)\boldsymbol{W}_1^{(1)} + t\boldsymbol{W}_1^{(2)}\right) \boldsymbol{X}_0,$$

$$\boldsymbol{l}_a(t) = \left((1-t)\boldsymbol{W}_L^{(1)} + t\boldsymbol{W}_L^{(2)}\boldsymbol{P}_{L-1}^T\right) \sigma \left((1-t)\boldsymbol{W}_{L-1}^{(1)} + t\boldsymbol{P}_{L-1}\boldsymbol{W}_{L-1}^{(2)}\boldsymbol{P}_{L-2}^T\right) \dots \tag{13}$$
$$\sigma \left((1-t)\boldsymbol{W}_1^{(1)} + t\boldsymbol{P}_1\boldsymbol{W}_1^{(2)}\right) \boldsymbol{X}_0.$$

For layer $i$ of networks along the interpolation, we define the pre-activations, $\boldsymbol{f}_i$, and post-activations, $\boldsymbol{g}_i$,

$$\boldsymbol{f}_1^u(t) = \left((1-t)\boldsymbol{W}_1^{(1)} + t\boldsymbol{W}_1^{(2)}\right) \boldsymbol{X}_0 \tag{14}$$
$$\boldsymbol{g}_i^u(t) = \sigma \boldsymbol{f}_i^u(t)$$
$$\boldsymbol{f}_i^u(t) = \left((1-t)\boldsymbol{W}_i^{(1)} + t\boldsymbol{W}_i^{(2)}\right) \boldsymbol{g}_{i-1}^u(t).$$

These are defined similarly for the interpolation of the aligned networks, where we denote them as $\boldsymbol{f}_i^a$ and $\boldsymbol{g}_i^a$.

Now we consider the $L_2$ distance between the first layer pre-activation distributions and the endpoint intermediate activation distributions. We define the following relevant $L_2$ distances:

$$\boldsymbol{d}_1^u(t;0) = ||\boldsymbol{f}_1^u(t) - \boldsymbol{f}_1^u(0)||_2 = t|| - \boldsymbol{f}_1^u(0) + \boldsymbol{f}_1^u(1)||_2 \tag{15}$$
$$\boldsymbol{d}_1^u(t;1) = ||\boldsymbol{f}_1^u(t) - \boldsymbol{f}_1^u(1)||_2 = (1-t)|| - \boldsymbol{f}_1^u(0) + \boldsymbol{f}_1^u(1)||_2 \tag{16}$$
$$\boldsymbol{d}_1^a(t;0) = ||\boldsymbol{f}_1^a(t) - \boldsymbol{f}_1^u(0)||_2 = t|| - \boldsymbol{f}_1^u(0) + \boldsymbol{P}_1\boldsymbol{f}_1^u(1)||_2 \tag{17}$$
$$\boldsymbol{d}_1^a(t;1) = ||\boldsymbol{f}_1^a(t) - \boldsymbol{P}_1\boldsymbol{f}_1^u(1)||_2 = (1-t)|| - \boldsymbol{f}_1^u(0) + \boldsymbol{P}_1\boldsymbol{f}_1^u(1)||_2 \tag{18}$$

Notice that $\boldsymbol{P}_1$ is the permutation associated with minimizing the ground metric $L_2$ norm for the Wasserstein distance between the endpoint models $\boldsymbol{f}_1^u(0)$ and $\boldsymbol{f}_1^u(1)$. Then it immediately follows that

$$\boldsymbol{d}_1^a(t;0) \leq \boldsymbol{d}_1^u(t;0) \qquad \boldsymbol{d}_1^a(t;1) \leq \boldsymbol{d}_1^u(t;1). \tag{19}$$

Thus, we have a tighter bound on the distance between the first layer pre-activations of models along the aligned curve than the unaligned curve to those of the endpoint models. We also have that the nonlinear pointwise activation function $\sigma$ is Lipschitz continuous. Thus, there exists a constant $L_\sigma$ such that

$$||\sigma \boldsymbol{f}_1^u(t) - \sigma \boldsymbol{f}_1^u(0)||_2 \leq L_\sigma \boldsymbol{d}_1^u(t;0). \tag{20}$$

Clearly, a similar relation holds for the other distances.

We calculate our distances $\boldsymbol{d}$ for the deeper layers of the network. We determine bounds on these distances, using $\boldsymbol{d}$ recursively, given in the following equations:

$$\boldsymbol{d}_i^u(t;0) = ||\boldsymbol{f}_i^u(t) - \boldsymbol{f}_i^u(0)||_2 \tag{21}$$
$$\leq L_\sigma ((1-t)||\boldsymbol{W}_i^{(1)}||_2 \boldsymbol{d}_{i-1}^u(t;0) + t||\boldsymbol{W}_i^{(2)}||_2 \boldsymbol{d}_{i-1}^u(t;1)) + t|| - \boldsymbol{f}_i^u(0) + \boldsymbol{f}_i^u(1)||_2$$
$$\boldsymbol{d}_i^u(t;1) = ||\boldsymbol{f}_i^u(t) - \boldsymbol{f}_i^u(1)||_2 \tag{22}$$
$$\leq L_\sigma ((1-t)||\boldsymbol{W}_i^{(1)}||_2 \boldsymbol{d}_{i-1}^u(t;0) + t||\boldsymbol{W}_i^{(2)}||_2 \boldsymbol{d}_{i-1}^u(t;1))$$
$$+ (1-t)|| - \boldsymbol{f}_i^u(0) + \boldsymbol{f}_i^u(1)||_2$$
$$\boldsymbol{d}_i^a(t;0) = ||\boldsymbol{f}_i^a(t) - \boldsymbol{f}_i^u(0)||_2 \tag{23}$$
$$\leq L_\sigma ((1-t)||\boldsymbol{W}_i^{(1)}||_2 \boldsymbol{d}_{i-1}^a(t;0) + t||\boldsymbol{W}_i^{(2)}||_2 \boldsymbol{d}_{i-1}^a(t;1)) + t|| - \boldsymbol{f}_i^u(0) + \boldsymbol{P}_i\boldsymbol{f}_i^u(1)||_2$$
$$\boldsymbol{d}_i^a(t;1) = ||\boldsymbol{f}_i^a(t) - \boldsymbol{P}_i\boldsymbol{f}_i^u(1)||_2 \tag{24}$$
$$\leq L_\sigma ((1-t)||\boldsymbol{W}_i^{(1)}||_2 \boldsymbol{d}_{i-1}^a(t;0) + t||\boldsymbol{W}_i^{(2)}||_2 \boldsymbol{d}_{i-1}^a(t;1))$$
$$+ (1-t)|| - \boldsymbol{f}_i^u(0) + \boldsymbol{P}_i\boldsymbol{f}_i^u(1)||_2$$

Using equation (19) and that $\boldsymbol{P}_i$ is chosen to minimize the $L_2$ distance of the intermediate pre-activations of the endpoints, it follows inductively that

$$\boldsymbol{d}_i^a(t;0) \leq \boldsymbol{d}_i^u(t;0) \qquad \boldsymbol{d}_i^a(t;1) \leq \boldsymbol{d}_i^u(t;1). \tag{25}$$

Thus, we have derived a tighter bound on the distance between the intermediate pre-activation distributions for models along the aligned linear interpolation to those of the endpoint.

Now, we make clear that the two endpoint networks are taken to be $\epsilon$ optimal networks. That is, $||\boldsymbol{Y} - \boldsymbol{Y}_j||_2 \leq \epsilon$, where $\boldsymbol{Y}$ are the true output for the training data. Clearly, given two trained networks such an $\epsilon$ must exist. This allows the following inequalities to hold,

$$\begin{aligned} ||\boldsymbol{l}_u(t) - \boldsymbol{Y}||_2 &\leq (1-t)||\boldsymbol{W}_L^{(1)}(\sigma\boldsymbol{f}_{L-1}^u(t) - \sigma\boldsymbol{f}_{L-1}^u(0))||_2 \\ &\quad + t||\boldsymbol{W}_L^{(2)}(\sigma\boldsymbol{f}_{L-1}^u(t) - \sigma\boldsymbol{f}_{L-1}^u(1))||_2 + \epsilon, \\ &\leq (1-t)||\boldsymbol{W}_L^{(1)}||_2 L_\sigma \boldsymbol{d}_{L-1}^u(t;0) + t||\boldsymbol{W}_L^{(2)}||_2 L_\sigma \boldsymbol{d}_{L-1}^u(t;1) + \epsilon. \end{aligned} \tag{26}$$

Similarly, we have that

$$||\boldsymbol{l}_a(t) - \boldsymbol{Y}||_2 \leq (1-t)||\boldsymbol{W}_L^{(1)}||_2 L_\sigma \boldsymbol{d}_{L-1}^a(t;0) + t||\boldsymbol{W}_L^{(2)}||_2 L_\sigma \boldsymbol{d}_{L-1}^a(t;1) + \epsilon. \tag{27}$$

Finally, we have that the loss function $\mathcal{L}$ is Lipschitz-continuous or that the input dataset is bounded. If only the later case is satisfied, then it follows that the output is also bounded. As the loss function is continuous, it follows that $\mathcal{L}$ is Lipschitz-continuous restricted to the image of the dataset under the neural networks. Then there exists a constant $L_L$ such that

$$\begin{aligned} \mathcal{L}(\boldsymbol{l}_u(t) - \boldsymbol{Y}) &\leq L_L ||\boldsymbol{l}_u(t) - \boldsymbol{Y}||_2 := B_u(t), \\ \mathcal{L}(\boldsymbol{l}_a(t) - \boldsymbol{Y}) &\leq L_L ||\boldsymbol{l}_a(t) - \boldsymbol{Y}||_2 := B_a(t). \end{aligned} \tag{28}$$

Now notice that since $\boldsymbol{d}_{L-1}^a(t) \leq \boldsymbol{d}_{L-1}^u(t)$, it follows that

$$B_a(t) \leq B_u(t), \quad t \in [0,1]. \tag{29}$$

Then we define upper bounds on the initializations for solving equation (5),

$$B_a := \int_0^1 B_a(t)dt \leq B_u := \int_0^1 B_u(t)dt, \tag{30}$$

where $\boldsymbol{P}$ is fixed as unaligned or determined by alignment. Thus, they are upper bounds for the optimal solutions. This completes the proof.

## C.2    On the Tightness of the Bounds

In the aforementioned proof, we derive tighter bounds for loss along the aligned curve compared to the unaligned curve, using the $L_2$ distance between pre-activations. We can establish the tightness of the provided bounds for a nontrivial class of networks. We do this to show that the bounds provide insight into how alignment aids mode connectivity, while having some practicality. The class of networks for which we will show tightness are networks with ReLU activation function and root mean squared error (RMSE) loss.

We emphasize that the bound depends on the following inequalities:

1. local Lipschitz continuity of the loss function
2. Lipschitz continuity of the activation function
3. matrix norm inequalities for the layer weights $\boldsymbol{W}_i$
4. triangle inequality for expressing the intermediate activations as a linear combination of those in the previous layer
5. triangle inequality related to $\epsilon$-optimality of the endpoints

First we show which weights allow the bounds for the linear interpolation between models to be tight.

1. Since the loss function is RMSE, equation (28) achieves equality with $L_L = 1$.

2. The ReLU activation function is Lipschitz continuous with $L_\sigma = 1$. This inequality is tight for bias vector large enough so that activations are non-negative.

3. The matrix norm inequality is met with equality when the weights $W_i$ act as isometries on the set of activations from the previous layer.

4. For these triangle inequalities to be met, all terms in the sum must have the same sign. This can be accomplished by a choice of bias vectors such that $\min f_i^u(1)$ is greater than $\max f_i^u(0)$. This is in addition to $W_i^{(1)} \geq 0$ and $W_i^{(2)} \leq 0$.

5. The final triangle inequality can be satisfied in the following way. The signs of the weight matrices can be found such that the endpoint activations have the same sign. The bias vectors can be found such that the endpoint activations are greater than the ground truth labels for some dataset. These choices of weights define a dataset for which the endpoint models are strictly $\epsilon_1$ and $\epsilon_2$ optimal respectively. Then $\epsilon$ in equation (27) can be replaced with the term, $(1-t)\epsilon_1 + t\epsilon_2$. These choices guarantee the last inequality is tight, albeit with a more specified epsilon.

Note that we show that there exists a choice of weights, $\epsilon_1$, and $\epsilon_2$ for which these bounds are tight.

In the main text, we discussed how tightness in the bounds for linear interpolation implies tightness in the bounds for continuous curves. Then, these bounds are nontrivial as we have tightness for a wide class of networks and curve parameterizations under a reasonable assumption.

### C.3 On the Use of Post-Activations for Neuron Alignment

In the previous proof in Appendix C.1, we assumed that the alignment is based on minimizing the $L_2$ distance of preactivations. The use of preactivation is needed in the calculation of $d_i$, where the term $f_i^u(t)$ can be linearly decomposed into $(1-t)W_i^{(1)}\sigma f_{i-1}^u(t)$ and $tW_i^{(2)}\sigma f_{i-1}^u(t)$. Such a decomposition does not necessarily hold for post-activations.

A natural question is how Theorem 3.1 can be applied to alignment of post-activations. This can be accomplished by modifying equation (2) in the following way. Let $C_{l,\text{pre}}$ and $C_{l,\text{post}}$ be the cost matrices of the $L_2$ distances of pre-activations and post-activations in network layer $l$ respectively. Then we can define the permutation associated with aligning post-activations as

$$P_l^* = \underset{P_l \in \Pi_{K_l}}{\arg\min} \operatorname{trace}(C_{l,\text{post}}^T P_l) \tag{31}$$

$$\text{such that} \quad \operatorname{trace}(C_{l,\text{pre}}^T P_l) \leq \operatorname{trace}(C_{l,\text{pre}}^T).$$

Given the added constraint, it follows that we can establish a tighter upper bound on the loss after aligning the post-activations, though this bound is not necessarily as tight as aligning preactivations. Using post-activations is more complicated theoretically due to the nonlinear nature of the activation function $\sigma$.

**On the Use of Cross-Correlation** In the main body of the paper, our numerical results concern the alignment given by maximizing the correlation of post-activations. We have just discussed theoretical details regarding the alignment of post-activations. Now, we address the use of cross-correlation. Given post-activations $g_i(0)$ and $g_i(1)$, if the distribution at each neuron is a unit normal Gaussian $\mathcal{N}(0, 1)$, then the alignment that maximizes cross-correlation is equivalent to the alignment that minimizes $L_2$ distance. In this sense, the use of cross-correlation approximates normalizing the distributions of post-activations before a $L_2$ minimizing alignment.

We provide an example to motivate the use of cross-correlation over unnormalized $L_2$ distance in our experiments. Consider a network with post-activations $g_i$ that will induce an alignment on the weights in the following linear layer of a neural network, $W_{i+1}$. With these quantities, we define quantities in what can be viewed as an equivalent network,

$$\hat{g}_i = \operatorname{diag}(\Sigma_{g_i})^{-1}(g_i - \mathbb{E}[g_i]), \tag{32}$$

$$\hat{W}_{i+1} = W_{i+1}(I + \mathbb{E}[g_i])\operatorname{diag}(\Sigma_{g_i}).$$

Note that these are pointwise normalizations of the post-activations. Then it is reasonable that if we were to align the activations $g_i$ and $\hat{g}_i$, we would want an alignment invariant to affine transformations

that can essentially be absorbed into the following linear layer. Additionally, maximizing cross-correlation as opposed to minimizing the $L_2$ distance of normalized distributions leads to the easy-to-interpret correlation signature seen in Figure 1.

In Figure 2, we compare these different techniques for alignment. Empirically, we find that aligning post-activations outperforms aligning pre-activations. Additionally, alignment via maximizing cross-correlation is seen to outperform all other methods. This validates are decision to use this technique in the main body of the paper.

## D  Residual Network Alignment

Algorithm 1 applies to networks with a typical feed-forward structure. In this section, we discuss how we compute alignments for the ResNet32 architecture as it is more complicated. It is important to align networks such that the network structure is preserved and network activations are not altered. In the context of residual networks, special consideration must be given to skip connections.

Consider the formulation of a basic skip connection,

$$\boldsymbol{X}_{k+1} = \sigma \circ (\boldsymbol{W}_{k+1}\boldsymbol{X}_k) + \boldsymbol{X}_{k-1} \tag{33}$$

In this equation, we can see that $\boldsymbol{X}_{k+1}$ and $\boldsymbol{X}_{k-1}$ share the same indexing of their units. This becomes clear when you consider permuting the hidden units in $\boldsymbol{X}_{k-1}$ without permuting the hidden units of $\boldsymbol{X}_{k+1}$. It is impossible to do so without breaking the structure of the equation above, where there is essentially the use of an identity mapping from $\boldsymbol{X}_{k-1}$ to $\boldsymbol{X}_{k+1}$. This effect that skip connections has on the symmetries of the loss surface has been studied previously in (Orhan & Pitkow, 2017). We note that the skip connection does not eliminate this symmetry, the symmetry is now just shared across certain layers.

We consider a traditional residual network that is decomposed into residual blocks. In each block the even layers have skip connections while the odd layers do not. So, we compute the alignment as usual for odd layers. For all even layers within a given residual block, we determine a shared alignment. We do this by solving the assignment problem for the average of the cross-correlation matrix over the even layers in that residual block.

## E  Proximal Alternating Minimization for Solving the Joint Model

We introduce a framework to solve the generalized problem in equation (5). Theoretically, this problem is fairly complicated and hard to analyze. Numerically, approaching the problem directly with first order methods could be computationally intensive as we need to store gradients of $\phi$ and $\boldsymbol{P}$ simultaneously. The problem can be more easily addressed using the method of proximal alternating minimization (PAM) (Attouch et al., 2010). The PAM scheme involves iteratively solving the two subproblems in equation (34). Here we let $Q(\phi, \boldsymbol{P})$ denote the objective function in equation (5). We only consider parameterized forms of $\boldsymbol{r}$ that satisfy the endpoint constraints for all $\phi$ and $\boldsymbol{P}$.

$$\begin{cases} \boldsymbol{P}^{k+1} = \underset{\boldsymbol{P}}{\arg\min} \quad Q(\phi^k, \boldsymbol{P}) + \frac{1}{2\nu_P}||\boldsymbol{P} - \boldsymbol{P}^k||_2^2 \\ \quad \text{such that} \quad \text{blockdiag}(\boldsymbol{P}_1, \boldsymbol{P}_2, ..., \boldsymbol{P}_{L-1}) \quad \text{where} \quad \boldsymbol{P}_l \in \Pi_{|K_l|} \\ \phi^{k+1} = \underset{\phi}{\arg\min} \quad Q(\phi, \boldsymbol{P}^{k+1}) + \frac{1}{2\nu_\phi}||\phi - \phi^k||_2^2 \end{cases} \tag{34}$$

Computing the unaligned curve is equivalent to solving the PAM scheme with a very small value of $\nu_P$. In fact, we are able to prove local convergence results for a certain class of networks.

**Theorem E.1** (Convergence). *Let $\{\phi^{k+1}, \boldsymbol{P}^{k+1}\}$ be the sequence produced by equation (34). Assume that $\boldsymbol{r}_\phi(t)$ corresponds to a feed-forward neural network with activation function $\sigma$ for $t \in [0, 1]$. Assume that $\mathcal{L}$, $r_\phi$, and $\sigma$ are all piece-wise analytic functions in $C^1$ and locally Lipschitz differentiable in $\phi$ and $\boldsymbol{P}$. Lastly, assume that the input data is bounded and the norm of the network weights are constrained to be bounded above. Then the following statements hold:*

1. $Q(\phi^{k+1}, \boldsymbol{P}^{k+1}) + \frac{1}{2\nu_\phi}||\phi^{k+1} - \phi^k||_2^2 + \frac{1}{2\nu_P}||\boldsymbol{P}^{k+1} - \boldsymbol{P}^k||_2^2 \leq Q(\phi^k, \boldsymbol{P}^k), \quad \forall k \geq 0$

2. $\{\phi^k, \boldsymbol{P}^k\}$ *converges to critical point of equation (5).*

*Proof.*  See Appendix E.3.

### E.1 Quadratic Bezier Curve Parameterization

We explicitly define the quadratic Bezier curve for use in the PAM algorithm in equation (35). Here the curve has been reparameterized so that the control point is a function of the permutation $\boldsymbol{P}$. $\tilde{\boldsymbol{\theta}}_c$ captures the deviation of the control point from the linear midpoint between $\boldsymbol{\theta}_1$ and $\boldsymbol{P}\boldsymbol{\theta}_2$. For PAM, $\tilde{\boldsymbol{\theta}}_c$ is the learnable curve parameter in $\phi$. It is zero initialized so that the initial curve is a linear interpolation between models as in traditional curve finding. This coupling of the control point with the permutation is critical for the success of PAM.

$$\boldsymbol{r}(t; \tilde{\boldsymbol{\theta}}_c, \boldsymbol{P}) = (1-t)^2\boldsymbol{\theta}_1 + t^2\boldsymbol{P}\boldsymbol{\theta}_2 + 2(1-t)t\left(\frac{\boldsymbol{\theta}_1 + \boldsymbol{P}\boldsymbol{\theta}_2}{2} + \tilde{\boldsymbol{\theta}}_c\right). \tag{35}$$

### E.2 Numerical Implementation for PAM

To learn each PAM curve, we perform a single outer iteration of PAM. This was seen as sufficient for training to converge. The permutation subproblem entails 20 epochs of projected stochastic gradient descent to the set of doubly stochastic matrices. This is done as the set of doubly stochastic matrices is the convex relaxation of the set of permutations. This projection is accomplished through 20 iterations of alternating projection of the updated permutation to the set of nonnegative matrices and the set of matrices with row and column sum of 1. After the 20 epochs of PGD, each layer permutation is projected to the set of permutations, $\Pi_{|K_l|}$. This projection is detailed in Appendix E.4. The curve parameter subproblem, which optimizes $\tilde{\boldsymbol{\theta}}_c$ from equation (35), entails 250 epochs of SGD. The same hyperparameters are used as in training the other curves. The learning rates are annealed with each iteration of PAM.

### E.3 Proofs for PAM

For the following proofs, we first establish and more rigorously define some terminology. We first discuss an important abuse of notation. For clarity the parameterized curve connecting networks under some permutation $\boldsymbol{P}$ that has been written as $\boldsymbol{r}_\phi(t)$ will now sometimes be referred to as $\boldsymbol{r}(t; \phi, \boldsymbol{P})$.

**Feed-forward neural networks**    In this section, we will be analyzing feed-forward neural networks. We let $\boldsymbol{X}_0 \in \mathbb{R}^{m_0 \times d}$ be the input to the neural network, $d$ samples of dimension $m_0$. Then we let $\boldsymbol{W}_i \in \mathbb{R}^{m_i \times m_{i-1}}$ denote the network weights mapping from layer $l-1$ to layer $l$. Additionally, $\sigma$ denotes the pointwise activation function. Then we can express the output of a feed-forward neural network, $\boldsymbol{Y}$, as:

$$\boldsymbol{Y} := \boldsymbol{W}_L\sigma \circ \boldsymbol{W}_{L-1}\sigma \circ \boldsymbol{W}_{L-2}...\sigma \circ \boldsymbol{W}_1\boldsymbol{X}_0 \tag{36}$$

To include biases, $\{\boldsymbol{b}_i\}_{i=1}^{L}$, we simply convert to homogeneous coordinates,

$$\hat{\boldsymbol{X}}_0 = \begin{bmatrix} \boldsymbol{X}_0 \\ 1 \end{bmatrix}, \quad \hat{\boldsymbol{W}}_i = \begin{bmatrix} \boldsymbol{W}_i & \boldsymbol{b}_i \\ \boldsymbol{0} & 1 \end{bmatrix}, \quad \hat{\boldsymbol{Y}} = \begin{bmatrix} \boldsymbol{Y} \\ 1 \end{bmatrix} \tag{37}$$

In all proofs, these terms are interchangeable.

**Huberized ReLU**    The commonly used ReLU function is defined as $\sigma(t) := \max(0, t)$. However, this function is not in $C^1$ and hence not locally Lipschitz differentiable. This makes conducting analysis with this function difficult. Thus, we will approach studying it through the lens of the huberized ReLU function, defined as:

$$\sigma_\delta(t) := \begin{cases} 0 & \text{for } t \leq 0 \\ \frac{1}{2\delta}t^2 & \text{for } 0 \leq t \leq \delta \\ t - \frac{\delta}{2} & \text{for } \delta \leq t \end{cases} \tag{38}$$

It is clear that $\sigma_\delta$ is a $C^1$ approximation of $\sigma$ such that $||\sigma - \sigma_\delta||_\infty = \frac{\delta}{2}$. Using huberized forms of loss functions for analysis is a fairly common technique such as in (Xu et al., 2016) which studies huberized support vector machines.

**Kurdyka-Lojasiewicz property** The function $f$ is said to have the Kurdyka-Lojasiewics (KL) property at $\bar{x}$ if there exist $\nu \in (0, +\infty]$, a neighborhood $U$ of $\bar{x}$ and a continuous concave function $\psi : [0, \nu) \to \mathbb{R}_+$ such that:

- $\psi(0) = 0$
- $\psi$ is $C^1$ on $(0, \nu)$
- $\forall s \in (0, \nu), \psi'(s) > 0$
- $\forall x \in U \cap [f(\bar{x}) < f < f(\bar{x}) + \nu]$, the Kurdyka-Lojasiewics inequality holds

$$\psi'(f(x) - f(\bar{x}))\text{dist}(0, \partial f(x)) \geq 1. \tag{39}$$

Here $\partial f$ denotes the subdifferential of $f$. Informally, a function that satisfies this inequality is one whose range can be reparameterized such that a kink occurs at its minimum. More intuitively, if $\psi$ has the form, $s^{1-\theta}$ with $\theta$ in $(0, 1)$, and $f$ is differentiable on $(0, \nu)$, then the inequality reduces to

$$\frac{1}{(1-\theta)}|f(x) - f(\bar{x})|^\theta \leq ||\nabla f(x)|| \tag{40}$$

**Semialgebraic function** A subset of $\mathbb{R}^n$ is semialgebraic if it can be written as a finite union of sets of the form

$$\{x \in \mathbb{R}^n : p_i(x) = 0, q_i(x) < 0, i = \{1, 2, ..., p\}\}$$

where $p_i$ and $q_i$ are real polynomial functions. A function $f : \mathbb{R}^n \to \mathbb{R} \cup \{+\infty\}$ is said to be semialgebraic if its graph is a semialgebraic subset of $\mathbb{R}^{n+1}$.

**Subanalytic function** Globally subanalytic sets are sets that can be obtained through finite intersections and finite unions of sets of the form $\{(x, t) \in [-1, 1]^n \times \mathbb{R} : f(x) = t\}$ where $f : [-1, 1]^n \to \mathbb{R}$ is an analytic function that can be extended analytically on a neighborhood of the interval $[-1, 1]^n$. A function is subanalytic if its graph is a globally subanalytic set.

### E.3.1 Proof of Theorem E.1

To prove this, we need that our problem meets the conditions required for local convergence of proximal alternating minimization (PAM) described in (Attouch et al., 2010). This requires the following:

1. Each term in the objective function containing only one primal variable is bounded below and lower semicontinuous.

2. Each term in the objective function which contains both variables is in $C^1$ and is locally Lipschitz differentiable.

3. The objective function satisfies the Kurdyka-Lojasiewicz (KL) property.

First we reformulate the problem so that it becomes unconstrained. Let $\chi$ denote the indicator function, where:

$$\chi_C(t) := \begin{cases} 0, & \text{for } t \in C \\ +\infty, & \text{otherwise} \end{cases} \tag{41}$$

This problem contains two hard constraints. First, each permutation matrix, $P_l$, must clearly be restricted to the set of permutation matrices of size $|K_l|$, $\Pi_{|K_l|}$. Additionally, it is assumed that the norm of the weights are bounded above. Without loss of generality, let $K_W$ denote an upper bound valid for all the weights. We denote the set of weights that satisfy the norm constraint as $\{A : ||A||_2^2 \leq K_W\}$. Then equation (5) with added regularization is equivalent to:

$$\phi^*, \boldsymbol{P}^* = \arg\min_{\phi, \boldsymbol{P}} \quad Q(\phi, \boldsymbol{P}) + \sum_{l=1}^{L-1} \chi_{\Pi_{|K_l|}}(\boldsymbol{P}_l) + \sum_{l=1}^{L} \chi_{\{A:||A||_2^2 < K_W\}}(W_l) \tag{42}$$

We now address each requirement for local convergence.

1. From equation (41), we can see that the sum of indicator functions are bounded below and lower semicontinuous.

2. Now we consider the form of the function, $Q(\phi, \boldsymbol{P})$. It has been defined as

$$\int_{t=0}^{1} \mathcal{L}(\boldsymbol{r}(t; \phi, \boldsymbol{P}))dt$$

We know that $\boldsymbol{r}(t; \phi, \boldsymbol{P})$ corresponds to a feed-forward neural network. Then $Q$ can be expressed as:

$$\int_{t=0}^{1} \mathcal{L}\left(W_L(t; \phi, \boldsymbol{P})\sigma \circ W_{L-1}(t; \phi, \boldsymbol{P})\ldots\sigma \circ W_1(t; \phi, \boldsymbol{P})X_0\right)dt \qquad (43)$$

with weight matrices $W_i$ and activation function $\sigma$. It becomes clear that for $Q(\phi, \boldsymbol{P})$ to be in $C^1$ and locally Lipschitz differentiable, the same must be true for $\mathcal{L}$, $\sigma$, and $\{W_i\}_{i=1}^{L}$. The first two are true as they are assumptions of the theorem. Since, $r_\phi$ is in $C^1$ and locally Lipschitz differentiable in the primal variables, then this is also true for all $W_i$. Thus, $Q(\phi, \boldsymbol{P})$ is in $C^1$ and locally Lipschitz differentiable.

3. To satisfy the KL property, the objective function must be a *tame* function (Attouch et al., 2010). Rigorously, this means that the graph of the function belongs to an o-minimal structure, a concept from algebraic geometry. We refer curious readers to (van den Dries & Speissegger, 2002) for further reference.

First, we note that $Q(\phi, \boldsymbol{P})$ is piece-wise analytic. This is because $Q$ is a composition of piece-wise analytic functions, $\mathcal{L}$, $\sigma$, and $r_\phi$. Additionally, because the input data is bounded and the norm of the weight matrices are bounded, it follows that the domain of $Q$ is bounded. Since, $Q$ is a piece-wise analytic function with bounded domain, it follows that $Q$ is a subanalytic function. The boundedness of the domain is an important detail here. This is because analytic functions are not necessarilly subanalytic unless their domain is bounded; a popular example of such a function is the exponential function.

We now consider the constraints associated with this problem, which have been re-expressed as indicator functions in the objective. The set of permutation matrices, $\Pi_{|K_l|}$, is finite and thus it is clearly a semi-algebraic set. Notice that the set of weight matrices satisfying the norm bound is equivalent to $\{A : ||A||_2^2 - K_W < 0\}$. The function that defines this set is a polynomial, so it is a semi-algebraic set. Indicator functions on semi-algebraic sets are semi-algebraic functions. Thus, the indicator functions in the objective are semi-algebraic.

The graphs of semi-algebraic functions and subanalytic functions both belong to the logarithmic-exponential structure, an o-minimal structure. A basic algebraic property of o-minimal structures is that the graphs of addition and multiplication are also elements of the structure (van den Dries & Speissegger, 2002). Since our objective function is a linear combination of semi-algebraic functions and subanalytic functions, it follows that the graph of our objective function is an element of the logarithmic-exponential structure. Therefore, our objective function is a *tame* function and it satisfies the KL property.

### E.3.2 Considering Rectified Networks

Theorem E.1 does not extend to the class of rectified networks. However, we are still interested in contructing a sequence of iterates $\{\phi^k, \boldsymbol{P}^k\}$ such that the objective value, $\mathbb{E}_{t \sim U}[\mathcal{L}(\boldsymbol{r}(t; \phi^k, \boldsymbol{P}^k))]$, is monotonic decreasing. The following theorem will introduce a technique for constructing such a sequence.

**Lemma E.2** ($\mathcal{L}$ restricted to possible network outputs is Lipschitz continuous). *For a feed-forward neural network, assume that $\mathcal{L}$ is continuous and that the neural network input, $X_0$, is bounded. Additionally, assume that the spectral norm of all weights, $\{W_i\}_{i=1}^{L}$, is bounded above by $K_W$, and the activation function, $\sigma$, is continous with $||\sigma|| \leq 1$. Let $S_Y$ denote the set of $Y$ where*

$$Y = \boldsymbol{W}_L \sigma \circ \boldsymbol{W}_{L-1} \sigma \circ \boldsymbol{W}_{L-2}...\sigma \circ \boldsymbol{W}_1 \boldsymbol{X}_0 \qquad (44)$$
$$such\ that \quad ||\boldsymbol{W}_i||_2 \leq K_W \quad \forall i \in \{1, 2, ..., L\}$$

*Then $\mathcal{L}$ restricted to the set $S_Y$ is Lipschitz continuous with some Lipschitz constant $K$.*

*Proof.* Since $X_0$ is bounded, it follows that there exists some constant $K_X$ such that $||X_0|| \leq K_X$. Since, the spectral norm of $W_1$ is bounded above by $K_W$, it is easy to see that $||W_1 X_0|| \leq K_W K_X$. Now since the pointwise activation function is a non-expansive map, it immediately follows that

$||\sigma \circ W_1 X_0|| \leq K_W K_X$. Following this process inductively, we see that the network output, $Y$, is bounded and that:

$$||Y|| \leq K_W^L K_X \qquad (45)$$

Since $Y$ is arbitrary, it follows that this is a bound for $S_Y$. Then we can restrict $\mathcal{L}$ to the ball in $\mathbb{R}^{m_L \times d}$ of radius $K_W^L K_X$. This ball is compact and $\mathcal{L}$ is continuous, so it follows that $\mathcal{L}$ restricted to this ball is Lipschitz continuous. Thus, there exists some Lipschitz constant $K$. Clearly, $S_Y$ is contained in this ball. Therefore, $\mathcal{L}$ is Lipschitz continuous on the set of all possible network outputs with Lipschitz constant $K$. □

Let $\theta_1$ and $\theta_2$ be feed-forward neural networks with ReLU activation function. Assume that $\mathcal{L}$ and $r_\phi$ are piece-wise analytic functions in $C^1$ and locally Lipschitz differentiable. Assume that the maximum network width at any layer is $M$ units. Additionally, assume that the network weights have a spectral norm bounded above by $K_W$, and that this is a hard constraint when training the networks. Finally, any point on $r_\phi$ must be equivalent to an affine combination of neural networks (Bezier curves, polygonal chains, etc.) satisfying the previously stated spectral norm bound.

Create the parameterized curve $\boldsymbol{r}_\delta(t; \phi, \boldsymbol{P})$ by substituting the huberized ReLU function, $\sigma_\delta$, into all ReLU functions in $\boldsymbol{r}(t; \phi, \boldsymbol{P})$. We refer to the objective values associated with these curves as $Q_\delta(\phi, \boldsymbol{P})$ and $Q(\phi, \boldsymbol{P})$ respectively.

**Theorem E.3** (Monotonic Decreasing Sequence for Rectified Networks). *For a feed-forward network, assume the above assumptions have been met. Additionally, assume that $X_0$ is bounded, so that $\mathcal{L}$ restricted to the set of possible network outputs is Lipschitz continuous with Lipschitz constant $K_L$ by Lemma E.2. Now generate the sequence $\{\phi^k, \boldsymbol{P}^k\}$ by solving equation (34) for $r_\delta(t; \phi, \boldsymbol{P})$. On this sequence impose the additional stopping criteria that*

$$\frac{1}{2\nu_\phi}||\phi^{k+1} - \phi^k||_2^2 + \frac{1}{2\nu_P}||\boldsymbol{P}^{k+1} - \boldsymbol{P}^k||_2^2 \geq K_L\sqrt{M}\frac{\delta}{2}\sum_{i=1}^{L-1}K_W^i \qquad \forall k \geq 0. \qquad (46)$$

*Then, the sequence of curves $\boldsymbol{r}(t; \phi^k, \boldsymbol{P}^k)$ connecting rectified networks has monotonic decreasing objective value.*

*Proof.* First we consider the approximation error from replacing $\sigma$ with $\sigma_\delta$. It is straightforward to see that

$$\max_t |\sigma(t) - \sigma_\delta(t)| \leq \frac{\delta}{2}. \qquad (47)$$

Then it follows that for any input $\boldsymbol{x}$,

$$||\sigma \circ W_1\boldsymbol{x} - \sigma_\delta \circ W_1\boldsymbol{x}||_2 \leq \sqrt{M}\frac{\delta}{2}.$$

Since the spectral norm of $W_i$ are bounded above by $K_W$, then we see that

$$||W_2\sigma \circ W_1\boldsymbol{x} - W_2\sigma_\delta \circ W_1\boldsymbol{x}||_2 \leq K_W\sqrt{M}\frac{\delta}{2}.$$

Now notice that

$$||\sigma \circ W_2\sigma \circ W_1\boldsymbol{x} - \sigma_\delta \circ W_2\sigma_\delta \circ W_1\boldsymbol{x}|| \leq ||\sigma \circ W_2\sigma \circ W_1\boldsymbol{x} - \sigma \circ W_2\sigma_\delta \circ W_1\boldsymbol{x}||$$
$$+ ||\sigma \circ W_2\sigma_\delta \circ W_1\boldsymbol{x} - \sigma_\delta \circ W_2\sigma_\delta \circ W_1\boldsymbol{x}||.$$

Since the ReLU function is a non-expansive map, it must be that the first term is bounded above by the previous error, $K_W\sqrt{M}\frac{\delta}{2}$. The second term corresponds once again to the error associated with the huberized form of the ReLU function, $\sqrt{M}\frac{\delta}{2}$. Thus the total error can be bounded by $(K_W + 1)\sqrt{M}\frac{\delta}{2}$.

Following this inductively, it can be seen that the this error grows geometrically with the number of layers. Additionally, the loss function is Lipschitz continuous when restricted to the set of possible network outputs. So we find the following bounds:

$$||Y - Y_\delta|| \leq \sqrt{M}\frac{\delta}{2}\sum_{i=1}^{L-1}K_W^i$$

$$||\mathcal{L}(Y) - \mathcal{L}(Y_\delta)|| \leq K_L\sqrt{M}\frac{\delta}{2}\sum_{i=1}^{L-1}K_W^i \qquad (48)$$

Since any point on the curve is an affine combination of networks with the $K_W$ bound on the spectral norm of their weights, it immediately follows this spectral norm bound also holds for the weights for any point on the curve. Then $||Q(\phi, \boldsymbol{P}) - Q_\delta(\phi, \boldsymbol{P})||$ is also bounded above by the bound in equation (48).

Then let $\{\phi^k, \boldsymbol{P}^k\}$ be the sequence generated by solving equation (34) using the curve $r_\delta$. $\sigma_\delta$ is a piece-wise analytic function in $C^1$ and is locally Lipschitz differentiable. Additionally, the spectral norm constraint on the weights is semi-algebraic and bounded below, so Theorem E.1 can be applied. It then follows that

$$
Q(\phi^{k+1}, \boldsymbol{P}^{k+1}) + \frac{1}{2\nu_\phi}||\phi^{k+1} - \phi^k||_2^2 + \frac{1}{2\nu_P}||\boldsymbol{P}^{k+1} - \boldsymbol{P}^k||_2^2
$$
$$
\leq Q(\phi^k, \boldsymbol{P}^k) + K_L\sqrt{M}\frac{\delta}{2}\sum_{i=1}^{L-1} K_W^i, \quad \forall k \geq 0 \tag{49}
$$

Thus, $\boldsymbol{r}(t; \phi^k, \boldsymbol{P}^k)$ is a sequence of curves, connecting rectified networks, with monotonic decreasing objective value as long as

$$
\frac{1}{2\nu_\phi}||\phi^{k+1} - \phi^k||_2^2 + \frac{1}{2\nu_P}||\boldsymbol{P}^{k+1} - \boldsymbol{P}^k||_2^2 \geq K_L\sqrt{M}\frac{\delta}{2}\sum_{i=1}^{L-1} K_W^i \qquad \forall k \geq 0
$$

Since the above equation is a stopping criterion introduced in the theorem statement, it follows that we have constructed a sequence of curves, connecting rectified networks, with monotonic decreasing objective value. $\qquad\square$

### E.4 Details on Solving the Permutation Subproblem in PAM

In this section, we provide additional details on the solution to the permutation subproblem in equation (34). For short-hand notation, we re-express the subproblem as

$$
\boldsymbol{P}^{k+1} = \underset{\boldsymbol{P} \in \text{blockdiag}(\boldsymbol{P}_1, \dots, \boldsymbol{P}_{L-1}); s.t. \boldsymbol{P}_i \in \Pi_{|K_i|}}{\arg\min} L(\boldsymbol{P}; \phi^k, \boldsymbol{P}^k) \tag{50}
$$

As described in Section E.2, when considering the permutation subproblem, we begin by solving the convex relaxation of the subproblem. To be clear, we solve for the locally optimal matrix $\boldsymbol{D}^*$, which is blockwise doubly-stochastic, minimizing $L$ in equation (50), using projected stochastic gradient descent.

Critical to solving the permutation subproblem is obtaining $\boldsymbol{P}^{k+1}$ given $\boldsymbol{D}^*$. We utilize $\boldsymbol{D}^*$ to determine a set of block permutation matrices, $S$, with $\boldsymbol{P}^{k+1}$ as an element. Specifically,

$$
\boldsymbol{P}^{k+1} := \underset{\boldsymbol{P} \in S}{\arg\min} \mathcal{L}(\boldsymbol{P}; \phi^k, \boldsymbol{P}^k). \tag{51}
$$

Intuitively, a natural candidate for $S$ is the projection of $\boldsymbol{D}^*$ to the set of permutations, $\Pi$. We refer to this permutation as $P_\Pi(\boldsymbol{D}^*)$. This solution is a classic heuristic for solving integer programs. This heuristic is precisely solving the convex relaxation of an integer program and then projecting the optimal solution back to the feasible set. In practice, this projection can be solved using the Hungarian algorithm. That is, the projection is the solution to

$$
P_\Pi(\boldsymbol{D}^*) := \underset{\boldsymbol{P} \in \Pi}{\arg\min} -\text{trace}(\boldsymbol{P}^T \boldsymbol{D}^*). \tag{52}
$$

As this projection is a heuristic, there is no guarantee that $P_\Pi(\boldsymbol{D}^*)$ is more optimal than the previous permutation, $\boldsymbol{P}^k$. To this end, we also let $\boldsymbol{P}^k$ be in $S$. This prevents us from having our loss increase after solving the subproblem.

It is also possible that there exists a permutation near $\boldsymbol{D}^*$ that is more optimal than $P_\Pi(\boldsymbol{D}^*)$. To this end, we are interested in randomly sampling such matrices. To do this, we will construct the block permutation matrix $\boldsymbol{R} := \text{blockdiag}(\boldsymbol{R}_1, \dots, \boldsymbol{R}_{L-1})$ where $\boldsymbol{R}_i$ is the permutation matrix corresponding to the $i$th layer and is sampled from a distribution $\Omega_i$. For more concise notation, we will say that $R \sim \Omega$. As there is no definitive way for sampling permutation matrices from doubly stochastic matrices, we detail how we construct the distributions $\{\Omega_l\}_{l=1}^{L-1}$ as follows.

Table 3: Average test accuracy of models along the learned curves trained with different hyperparameters. The curves connect the TinyTen architecture and are trained on the CIFAR100 dataset. The accuracy of any choice of aligned curve exceeds that of all unaligned curves. This establishes that the performance gains associated with alignment are not sensitive to choice of batch size or intial learning rate.

| | Unaligned curves Batch size | | | Aligned curves Batch size | | |
|---|---|---|---|---|---|---|
| Learning rate | 64 | 128 | 256 | 64 | 128 | 256 |
| 1E-2 | $55.9 \pm 0.2$ | $55.3 \pm 0.3$ | $54.2 \pm 0.3$ | $58.4 \pm 0.2$ | $57.9 \pm 0.3$ | $57.4 \pm 0.1$ |
| 1E-1 | $55.9 \pm 0.1$ | $56.0 \pm 0.2$ | $56.0 \pm 0.1$ | $59.0 \pm 0.1$ | $58.7 \pm 0.2$ | $58.7 \pm 0.1$ |
| 5E-1 | $55.5 \pm 0.3$ | $55.7 \pm 0.3$ | $55.5 \pm 0.1$ | $58.6 \pm 0.2$ | $58.7 \pm 0.1$ | $58.9 \pm 0.1$ |

A well-known result is that every doubly stochastic matrix is a convex combination of permutation matrices, with this convex combination being known as a Birkhoff-von Neumann (BvN) decomposition of the matrix (Dufossé & Uçar, 2016). That is, for any doubly stochastic matrix $D$, there exists a set of permutation matrices such that

$$\boldsymbol{D} = \sum_{i \in I} \alpha_i \boldsymbol{P}_i \quad \text{s.t.} \quad \alpha \geq 0, \sum_{i \in I} \alpha_i = 1, \boldsymbol{P}_i \in \Pi. \tag{53}$$

The problem of determining the minimal size of the index set $I$ is known to be an NP hard problem, and a doubly stochastic matrix can have multiple BvN decompositions. For sampling the permutation from layer $l$, $\boldsymbol{P}_l$, we utilize a BvN decomposition. This is because the BvN decomposition lends itself to having a probabilistic interpretation. Given the decomposition in equation (53), we view the permutation $\boldsymbol{P}_i$ being sampled from $\Omega_l$ with probability $\alpha_i$.

To determine the specific BvN decomposition that we use, we introduce a variant of the *greedy Birkhoff heuristic* (Dufossé & Uçar, 2016). First, let $\boldsymbol{D}^{(1)} := \boldsymbol{D}$. We associate a bipartite graph, $G^{(1)}$, with the matrix. In this bipartite graph, the two vertex sets correspond to the rows and columns of $\boldsymbol{D}^{(1)}$ with an edge indicating a nonzero value in the corresponding entry of $\boldsymbol{D}^{(1)}$. We then consider the set of permutation matrices corresponding to a perfect matching in $G$, $\Pi_M$. With this, we define the first permutation in the BvN decomposition as

$$\boldsymbol{P}_1 = \arg\max_{\boldsymbol{P} \in \Pi_M} \text{trace}(\boldsymbol{P}^T \boldsymbol{D}^{(1)}). \tag{54}$$

Then $\alpha_1$ is taken to be the smallest nonzero term in $\boldsymbol{P}^T \boldsymbol{D}^{(1)}$. Following this, we take $\boldsymbol{D}^{(2)} = \boldsymbol{D}^{(1)} - \alpha_1 \boldsymbol{P}_1$ and iteratively repeat this process until the full BvN decomposition has been constructed.

This BvN decomposition can easily be solved iteratively using the Hungarian algorithm. Note that $P_\Pi(\boldsymbol{D}_l^*)$ is guaranteed to be the first term of the BvN decomposition with the highest value of $\alpha_1$ possible among BvN decompositions including that permutation. This motivates the use of this heuristic, as it can be seen as a natural probabilistic extension of direct projection to the permutation set. The traditional *greedy Birkhoff heuristic* determines $\boldsymbol{P}_l$ in the following way instead,

$$\boldsymbol{P}_l = \arg\max_{\boldsymbol{P} \in \Pi_M} \min \text{diag}(\boldsymbol{P}^T \boldsymbol{D}^{(1)}). \tag{55}$$

In practice, we truncate $\alpha$ after $i = 10$, and normalize the truncated coefficient vector to determine the distribution $\Omega_l$. We sample $M$ block permutation matrices $\boldsymbol{R}$ from $\Omega$, with $M = 32$. This is clearly a small sample given the dimension of $P$, but it is large enough to provide matrices that outperform $P_\Pi(\boldsymbol{D}^*)$ in our experiments.

Then all together, we have that the set $S$ contains the block permutation matrices, $\{P^k, P_\Pi(\boldsymbol{D}^*), \{\boldsymbol{R}^{(t)}\}_{t=1}^M; \boldsymbol{R}^{(t)} \sim \Omega\}$.

## F  Hyperparameter Search

In the main paper, the CIFAR100 curves are trained with an initial learning rate of 1E-1 and a batch size of 128. The choice of batch size is the same as in (Garipov et al., 2018) while training was seen

to converge with the given initial learning rate. Still, it is important to establish that are main results are not dependent on the choice of hyperparameters.

In Table 3, the results for training curves for the TinyTen architecture and CIFAR100 dataset using three different choices of initial learning rate and batch size are displayed. We see that the curves are mostly optimal when the choice of batch size is not too large and the learning rate is not too small. Regardless, it is clear that the accuracies associated with the aligned curves exceed those of the unaligned curves. Namely, the lowest of the aligned accuracies is notably greater than the highest of the unaligned accuracies. Thus, we can conclude that alignment improves mode connectivity while not being sensitive to the choice of hyperparameters. We believe this insensitivity to hyperparameters will extend to different architectures and datasets.