[Reviews · NeurIPS 2020]

Review 1

Summary and Contributions: This paper proposed Neuron Alignment, which is a heuristic method aiming to find a lower-loss curve between local minima on the loss landscape of neural networks by taking the symmetry of neurons into account. Specifically, Neuron Alignment tries to find a permutation of neurons in neural networks so that the corresponding weights of the two models in parameter space are more correlated, and then connect one of the minima with the permuted version of the other using existing methods. The authors theoretically proved that their method can decrease the upper bound of the expected loss on the curve, and empirically verified on various models and datasets that their proposed method finds lower-loss curves than the baseline. This method is also empirically proven to be locally optimal, and it can reduce the robust loss barrier on the curve between adversarially robust models.

Strengths: -The idea of "Neuron Alignment via Assignment", which takes neuron symmetry into account, is novel and interesting. Finding the best permutation of neurons to optimize the correlations can be hard because of the discrete nature of this problem, but the authors solved this problem by using heuristics and showed empirically that their solution is locally optimal. -The authors did various experiments and showed that their Neuron Alignment method can help to find a better path connecting the two local minima, where the min/avg accuracy along the path can be decreased by up to 3-4 percent. This phenomenon is consistent for different models (TinyTen/ResNet32/GoogLeNet) on different datasets (CIFAR-10/100, ImageNet).

Weaknesses: -The theoretical insight (Theorem 3.1) of Neuron Alignment is a bit weak. The authors only proved that the upper bound for the averaged loss decreases, but didn't talk about the tightness of this upper bound. Besides, there is a small gap between theory and experiments because the authors are using pre-activations for neuron alignment and the assumptions that the loss and activation function are Lipschitz-continuous, while in experiments they are using post-activations for neuron alignment and ReLU activations. -The Neuron Alignment algorithm heavily depends on the structure of the neural networks, e.g., the symmetry for neurons is different for convolutional/fully-connected layers and residual blocks. This may make the application of this algorithm somewhat limited.

Correctness: The theoretical claims and empirical methodology appear to be correct.

Clarity: This paper is generally well-written and well-structured with some minor problems: -In equation 2, the authors are talking about the correlations of activations without stating the definition of correlations. Since this is an important concept in this paper, it may be better if the authors could state the definition (or at least some explanation) of correlation in this paper instead of referring the readers to another paper. -When the authors are mentioning some tables and figures, it would be better if the authors could just say that these tables and figures are in the appendix so that the readers wouldn't waste time finding these things in the main paper. Examples include "Table 2" at line 223, "Figure 7" at line 265, and "Figure 8" at line 270. -For Figure 2 top left, the pictures are not clear enough and it's hard to see the texts like "\theta_1" in the figure, so perhaps the authors could find a better way of plotting this. -Typo: Line 265, "in in" -> "in"

Relation to Prior Work: This paper is related to previous works about mode connectivity and neuron symmetry/network similarity, all of which are discussed in this paper. The authors also clearly stated that their work bridges mode connectivity and neuron symmetry by introducing a heuristic method, i.e., Neuron Alignment, to find better curves for mode connectivity.

Reproducibility: Yes

Additional Feedback: I have read the other reviews and the authors' feedback, and I have decided to maintain my score. Generally, the authors addressed some of my concerns, but I am still worried about the tightness of the bound provided in Theorem 3.1, which is also my main concern. The detailed reasons why I keep my score are listed below: -Thank the authors for providing such detailed feedback! I have learned some more intuitions about the proof of Theorem 3.1. However, the proof still requires a lot of inequalities, e.g., triangle inequalities, matrix norm inequalities, and Lipschitz continuities. Some of the bounds can be tight for some specific network structures, but it's still a bit hard for me to believe the upper bound provided in this theorem is tight enough. I still think that an empirical computation of these bounds should be much more convincing than the current theoretical explanation given by the authors. -The authors did not address my concerns about the clarity of this paper. I think it is very important for a paper to be clearly written, but the current version of this paper has some clarity issues. -There is another paper called "Low-loss connection of weight vectors: distribution-based approaches", which I think is very related to this paper but was published after this paper was submitted to NeurIPS. It would be better if the authors could compare with that paper in the camera-ready version. Below is the original review. -------------------------------------------------- -For the upper bounds provided in Theorem 3.1, I wonder whether it is possible to empirically compute or approximate them (perhaps for some small networks with small datasets) so that one can have a sense of whether the inequalities are tight or not. It would also be better if the authors could provide more intuitions about this upper bound, e.g., how the upper bound is constructed. If the upper bounds are very loose, then the intuition provided by this theorem will be quite limited. -In this paper, the authors are permuting the orders of the neurons once. However, if we consider the weights of neural networks as a distribution, the order of them can be permuted many times along the optimal path. In other words, after traveling some distance from one local minima to the other, it may have another permutation that makes its weights more correlated to the destination. Thus, one possible future work might be interesting to investigate the possibility of "multi-stage" neuron alignments.


Review 2

Summary and Contributions: This paper focuses on the problem of curve finding, i.e. finding a curve connecting two modes (e.g. global minima) in a neural network loss landscape such that the average loss along the curve is as low as possible. The main contributions of this paper are: -- It proposes neuron alignment---a procedure that permutes a network in the parameter space but does not change it in the function space---as an addition to curve finding algorithms. The neuron alignment can either work as a modular pre-processing step to any curve finding algorithm, or be jointly optimized along with the curve. -- The paper shows experimentally that making the networks aligned can improve upon the performance of common curve finding algorithms. The same observation holds on connecting adversarially robust models. -- Some theoretical justification on why aligned networks may be better connected than unaligned networks.

Strengths: -- The idea of using neuron alignment (as a reparametrization) to enhance mode connectivity is pretty neat, and novel (as far as I’m aware of). It makes a lot of sense that permuting the neurons do not change one network, but can improve the connectivity in the parameter space when we use it to align two networks. -- The experiments seem comprehensive (3 networks of different sizes on 3 image classification tasks) and convincing enough to me. Also it’s good to see the same results hold on adversarially robust models. It’s a bit concerning that baseline curve finding already works reasonably well in terms of average loss (and thus the room for improvement using alignment is also little, cf. Table 1). I’m curious have the authors tried vanilla *linear* mode connectivity between aligned vs. unaligned networks? Because the curve is linear and not learnable, the performance will degrade for both aligned and unaligned, but I wonder if neuron alignment still helps there. -- At a high level, the experimental results provide us with a bit of new knowledge about mode connectivity / loss landscape of neural networks. -- I liked the proposal and discussion about joint optimization of permutation and curve (PAM). Especially the observation that PAM does not help when already initialized from a per-learned alignment, but still helps when initialized from identity permutation (i.e. no permutation). This suggests PAM sort of works but the simpler two-stage algorithm (alignment then curve learning) performs just as well so that maybe there is no need to run the more sophisticated PAM.

Weaknesses: -- I’m concerned about the strength / implications of the theoretical result (Theorem 3.1). Most importantly, the theorem states a comparison between unaligned and aligned networks in terms of curve finding, but the comparison is done on *upper bounds*, not actual values. In this case, it’s often good to have more explicit expressions for the two upper bounds (e.g. how they depend on various problem parameters), and discussions on how tight the bounds are, which are all missing in the current paper. Without these details, it could a priori be the case that both bounds are very loose, in which case it does not make sense to draw any conclusion about the two algorithms based on them. -- Algorithmic novelty. At the core the proposed algorithm seems to be a combination of existing techniques (learning permutations and curve finding). Because this is an understanding paper (in my perspective) though, I am probably weighing less on this point.

Correctness: The theoretical results as well as the experimental methodologies are correct up to my inspections.

Clarity: The paper is generally quite well presented and I didn’t have a hard time understanding it.

Relation to Prior Work: The paper discusses related work on mode connectivity and curve finding, weight symmetry, and network similarity. These seem all relevant enough and positions the present paper well in terms of its delta with prior work. I’m not very familiar though with the prior work in the specific directions of curve finding / mode connectivity, so may not have the best judgement in whether the related work is comprehensive enough.

Reproducibility: Yes

Additional Feedback:


Review 3

Summary and Contributions: The paper explores the loss landscape of trained neural nets. It focuses on the problem of mode connectivity. Given two optimized instances of the same neural net architecture, one seeks a curve on the loss landscape parametrized by their weights that connects the two. The authors continue the recent line of work, which learns curves with minimal impact to loss. The main observation is that several points in the weights space correspond to the same network due to weight symmetry. The authors continue to develop an algorithm that finds a locally suitable permutation using neuron alignment. Notably, the authors find robust models with a lower loss on the curve between two robust models. Update: I thank the authors' for their explanation. It would be good to add the subset size explanation to the final version. The score stays as it is.

Strengths: The main contribution to the NeurIPS community is an algorithm that finds a robust, optimized model with minimal cost. Another contribution is an insight into how important is exploiting symmetries for mode connectivity. The authors test their curve loss empirically and verify that it is locally optimal with Proximal Alternating minimization.

Weaknesses: Theorem 3.1, which shows tighter loss bounds, is weak in that it doesn't connect the aligned and unaligned losses. Algorithm 1 discussion should contain the impact of choosing a specific subset of X on the algorithm performance. The authors do not compare the resulting loss of neuron alignment to curves found by previous methods, e.g., Garipov et al.

Correctness: The methods and claims appear to be fully supported and correct: symmetry is theoretically and empirically shown to be an essential factor in curve optimization.

Clarity: The paper is well written. However, some parts of the text are a little long and vague- e.g. "Theory for using Neuron Alignment" paragraph, and better be shortened.

Relation to Prior Work: Yes, The authors discuss the importance of weight symmetry and compare it to many recent works of curve optimization in the related work section.

Reproducibility: Yes

Additional Feedback:

[Author Response · NeurIPS 2020]

**Tightness of bounds in Theorem 3.1 (All reviewers)** First we note that there is a nontrivial class of networks for which the upper bounds are tight when considering their linear interpolations. Namely, networks with ReLU activation function and mean squared error (MSE) loss. Section C.1 (supp. material) details the proof of Theorem 3.1. The bound depends on the following inequalities; Lipschitz continuity of the activation function, matrix norm inequalities for the layer weights $W_i$, a triangle inequality when expressing the intermediate activations in layer $i$ as the combination of those in layer $i-1$, a triangle inequality related to the $\epsilon$ optimality of the endpoints, and the local Lipschitz continuity of the loss function. For the class of networks mentioned, the last inequality becomes trivial given we have MSE loss. Weights can be found for ReLU networks such that the other inequalities are tight.

We now address the tightness of the bound for piecewise linear curves between networks, which themselves can approximate continuous curves. Under the assumption that a piecewise linear curve or limit of such sequence truly does exist along which the loss is optimal and constant (reasonable as mode connectivity assumes this), for two models on the same line segment of this curve, we can consider the optimal curve between them. It follows that this optimal curve is their linear interpolation. Therefore, we have tightness in the upper bound for curve-finding restricted to piecewise linear curves between a class of networks for which we have tightness in the linear interpolation for that class. Via an argument by continuity, this tightness extends to continuous curves. Thus, these bounds are nontrivial as we have tightness for a wide class of networks and curve parameterizations under a reasonable assumption. We will add the discussion on tightness in the revised version.

**Small gap between theory and experiments (R1)** We discuss the use of post-activations in section C.2. From Figure 9 (supp. material), it is clear the curve learned with alignment via pre-activations outperforms the unaligned curve. Thus, the theorem applies to a successful method. Since we see best performance from alignment using the correlation of post-activations, we use that method for the majority of the experiments. We discuss the relation between these two techniques in section C.2. To the reviewer's other point, we also note that ReLU is Lipschitz continuous with constant 1.

**Neuron Alignment algorithm and symmetry (R1)** The symmetry for dense layers, convolutional layers, and residual blocks do have different properties as the reviewer mentions. By conducting experiments on TinyTen, ResNet32, and GoogLeNet, we show that mode connectivity with neuron alignment is able to offer performance gains with these different layers present. We note that Figure 1a, shows that there is still meaningfully symmetry detected by neuron alignment for all of these mentioned layers. Thus, we believe the algorithm can be widely applied.

**Sketch of the proof of Theorem 3.1 (R1)** We can approximate the intermediate activations of a network along the linear interpolation as an interpolation of the endpoint activations. Then the distance between the activations to those of an endpoint is bounded by the sum of this approximation error and the distance between the two endpoint activations. Thus for aligned networks, this second term in that bound is smaller as the permutation minimizes that value. As we iteratively do this for each layer, we finally bound the distance between the output of the network along the curve to that of the endpoint models, giving us a bound on the error of the network that is tighter with alignment.

**Multi-stage alignment (R1)** We appreciate the suggestion of a multi-stage neuron alignment. We were curious if models along the curve are aligned to the endpoint models. Figures 1b and 1c show this for the curve midpoint. Interestingly, for aligned curves, the midpoint on the learned curve is already optimally aligned to the endpoints.

**Significance of performance (R3)** We emphasize our contribution by including the minimal accuracy along the curve in Table 1. We can see that there is a model along the unaligned curves that clearly has subpar performance compared to independently trained models (i.e. the endpoints). Alignment notably improves this. Additionally, we have included figures that show the performance of the linear mode connectivity with and without alignment. This can be seen in Figure 2 (top left) and Figure 4 (supp. material). The performance gain from alignment is very notable, up to 60% difference in minimal accuracy along the curve, in these figures.

**Expression of the bounds (R3)** The bounds can be stated recursively via the equations in section C.1. We chose to omit it from the main body in the interest of space. $B_u$ is a nonnegative linear combination of the endpoint activation distances, $||f_i^u(0) - f_i^u(1)||_2$. For $B_a$, the only difference in the expression is the use of $||f_i^u(0) - P_i f_i^u(1)||_2$ for each layer. So when the intermediate activations are closer, as in aligned networks, the upper bound becomes smaller.

**Comparisons to Garipov et al. (R4)** It seems that there was a misunderstanding. We compare the performance of aligned curves to unaligned curves in our results. When the curves are unaligned, these are precisely curves trained via the method of Garipov et al. Thus our paper shows improvement over existing methods.

**Choice of subset for Algorithm 1 (R4)** By a subset of the training data, we were referring to a random subset of the training data that well approximates the underlying data manifold. This is to emphasize that the alignment can be computed more quickly on a random subset compared to the whole training dataset. As a test, we computed the alignment for a pair of TinyTen networks trained on CIFAR100, where the subsets contained 2500, 5000, and 10000 images respectively. Despite these being random subsets of differing size, they produced the same alignment.

[Meta-Review · NeurIPS 2020]

This paper studies the mode connectivity phenomenon of neural network and tries improve mode connectivity using a technique called neuron alignment. The paper demonstrated (both theoretically and empirically) that neuron alignment can improve mode connectivity. While the reviewers all find the paper to be interesting, there were also some concerns that the theoretical result is focused on an upperbound of the loss and does not prove that the actual expected loss improves.